# GIT-BO: High-Dimensional Bayesian Optimization using Tabular Foundation Models

**Rosen Ting-Ying Yu, Cyril Picard, Faez Ahmed**
Massachusetts Institute of Technology
Massachusetts, MA 02139, USA
`{rosenyu, cyrilp, faez}@mit.edu`

## Abstract

Bayesian optimization (BO) struggles in high dimensions, where Gaussian-process surrogates demand heavy retraining and brittle assumptions, slowing progress on real engineering and design problems. We introduce GIT-BO, a Gradient-Informed BO framework that couples TabPFN v2, a tabular foundation model (TFM) that performs zero-shot Bayesian inference in context, with an active-subspace mechanism computed from the model's own predictive-mean gradients. This aligns exploration to an intrinsic low-dimensional subspace via a Fisher-information estimate and selects queries with a UCB acquisition, requiring no online retraining. Across 60 problem variants spanning 20 benchmarks—nine scalable synthetic families and ten real-world tasks (e.g., power systems, Rover, MOPTA08, Mazda)—up to 500 dimensions, GIT-BO delivers a stronger performance–time trade-off than state-of-the-art GP-based methods (SAASBO, TuRBO, Vanilla BO, BAxUS), ranking highest in performance and with runtime advantages that grow with dimensionality. Limitations include memory footprint and dependence on the capacity of the underlying TFM.

## 1 Introduction

Optimizing expensive black-box functions is central to progress in areas such as machine learning (Dewancker et al., 2016; Snoek et al., 2012), engineering design (Kumar et al., 2024; Wang & Dowling, 2022; Zhang et al., 2020; Yu et al., 2025), and hyperparameter tuning (Klein et al., 2017; Wu et al., 2019). Bayesian optimization (BO) has become the method of choice in these settings due to its sample efficiency. Yet, despite its successes, standard BO with Gaussian processes (GPs) is widely viewed as limited to low-dimensional regimes, typically fewer than a few dozen variables (Liu et al., 2020; Wang et al., 2023; Santoni et al., 2024). Scaling BO to hundreds of dimensions remains a critical barrier, where the curse of dimensionality, GP training costs, and hyperparameter sensitivity severely hinder performance. Research has sought to overcome these challenges through three main strategies: (1) Exploiting intrinsic low-dimensional structure, e.g., random embeddings and active subspaces (Wang et al., 2016; Nayebi et al., 2019; Letham et al., 2020; Papenmeier et al., 2022);(2) Additive functional decompositions (Kandasamy et al., 2015; Gardner et al., 2017; Rolland et al., 2018; Han et al., 2021; Ziomek & Ammar, 2023); and (3) Alternative GP priors and trust-region heuristics (Eriksson et al., 2019; Eriksson & Jankowiak, 2021; Hvarfner et al., 2024; Xu et al., 2025). These innovations push the frontier but still face two practical obstacles: (1) prohibitive computation from iterative GP retraining, and (2) brittle reliance on hyperparameter tuning, including determining appropriate intrinsic dimensionality and selecting optimal kernels and priors (Rana et al., 2017; Letham et al., 2020; Eriksson & Jankowiak, 2021).

Recent advances in tabular foundation models (TFMs) provide a radically different surrogate modeling paradigm. Prior-Data Fitted Networks (PFNs) (Müller et al., 2022; Hollmann et al., 2022; Müller et al., 2023) perform Bayesian inference in-context with frozen weights, eliminating kernel re-fitting and delivering 10–100x speedups on BO tasks (Rakotoarison et al., 2024; Yu et al., 2025). These approaches address computational bottlenecks by leveraging pre-trained models' in-context learning capability, which requires only a single forward pass at inference during optimization. These powerful TFMs trained on millions of synthetic prior data can also perform accurate inference without additional hyperparameter tuning for a new domain.

The newly released TabPFN v2 (Hollmann et al., 2025) extends this capacity to inputs up to 500 dimensions, opening the door to foundation-model surrogates for high-dimensional BO for the first time. However, prior analyses have shown that TabPFN v2, despite its strong performance on small-to medium-scale tasks, exhibits performance degradation in high-dimensional regimes, with recent work proposing divide-and-conquer or feature-extraction strategies to mitigate these limitations (Ye et al., 2025; Reuter et al., 2025). This raises a fundamental question: *Are frozen TFMs sufficient for high-dimensional optimization, or must they be coupled with classical algorithmic strategies to succeed?*

We answer this question by introducing Gradient-Informed Bayesian Optimization using TabPFN (GIT-BO), a framework that integrates TabPFN v2 with gradient-informed active subspaces. Our key idea is to exploit predictive-mean gradients available from the frozen model itself to construct low-dimensional gradient-informed subspaces. This provides algorithmic guidance for adaptive exploration while preserving the inference-time efficiency of TFMs. In doing so, GIT-BO aligns foundation models with classical subspace discovery, combining the speed of in-context surrogates with the structural power needed in extreme dimensions. We perform comprehensive algorithm benchmarking against the state-of-the-art (SOTA) GPU-accelerated high-dimensional BO algorithms and test them on commonly used synthetic benchmarks as well as several real-world engineering BO benchmarks.

Our contributions are:

- We propose GIT-BO, a gradient-informed high-dimensional BO method that adaptively discovers active subspaces from a frozen foundation model's predictive gradients, requiring no online retraining.
- Across 60 diverse problems comprising synthetic and real-world benchmarks (including power systems, car crash, and dynamics control), GIT-BO consistently achieves state-of-the-art optimization quality with orders-of-magnitude runtime savings compared to GP-based methods.

Our results demonstrate that foundation model surrogates, when paired with structural algorithmic guidance, emerge as viable and competitive alternatives to Gaussian-process-based BO for high-dimensional problems.

## 2 BACKGROUND

### 2.1 HIGH-DIMENSIONAL BAYESIAN OPTIMIZATION

Bayesian optimization is a sample-efficient approach for optimizing expensive black-box functions where the objective is to find $x^* \in \arg\max_{x \in \mathcal{X}} f(x)$ with $\mathcal{X} = [0, 1]^D$, achieved by sequentially querying promising points under the guidance of a surrogate model. Gaussian processes (GPs) remain the dominant surrogate due to their effective uncertainty-based exploration and exploitation, but their cubical computational scaling and deteriorating performance with increasing dimensionality pose serious challenges (Liu et al., 2020; Wang et al., 2023; Santoni et al., 2024; Ramchandran et al., 2025). Three main families address these issues:

**Exploiting intrinsic low-dimensional structure.** A common strategy in high-dimensional BO is to assume the objective depends on only a few effective directions and to project the search into that subspace, where GPs perform more reliably. REMBO introduced random linear projections (Wang et al., 2016), while HESBO (Nayebi et al., 2019) and ALEBO (Letham et al., 2020) refined this idea using sparse embeddings and Mahalanobis kernels. More recently BAxUS (Papenmeier et al., 2022), adaptively expands nested subspaces with guarantees. These succeed when a meaningful active subspace exists, but degrade when structure is weak or mis-specified.

**Additive decompositions.** Another approach assumes the objective decomposes into a sum of low-dimensional components, enabling separate GP models. Additive GPs use disjoint decompositions (Kandasamy et al., 2015), while later work allows overlaps (Rolland et al., 2018) or tree-structured dependencies (Han et al., 2021) to improve tractability. Randomized decompositions offer a lightweight alternative (Ziomek & Ammar, 2023). These methods improve sample efficiency

since each component is easier to model, but remain limited by the difficulty of discovering the right decomposition from sparse data and the overhead of structure learning, restricting adoption in practice (Rolland et al., 2018; Han et al., 2021; Ziomek & Ammar, 2023).

**Alternative modeling and trust-region strategies.** Beyond embeddings and additive decompositions, another line of work rethinks the surrogate itself. SAASBO introduces sparsity-inducing shrinkage priors on GP length-scales to identify relevant dimensions automatically (Eriksson & Jankowiak, 2021).TuRBO (Eriksson et al., 2019) replaces global modeling in favor of multiple local GP surrogates confined to dynamically adjusted trust regions. More recently, studies show that vanilla BO with carefully chosen priors (Hvarfner et al., 2024) and standard GPs with robust Matérn kernels (Xu et al., 2025) can remain competitive in high dimensions.

Despite these advances, such methods still depend on high-to-low-dimensional learning, sensitive kernel choices, or strong structural assumptions—motivating foundation model surrogates as a fundamentally different path forward.

## 2.2 Tabular Foundation Models as BO Surrogates

Tabular foundation models (TFMs) provide amortized Bayesian inference through in-context learning (ICL). Prior-Data Fitted Networks (Müller et al., 2022; Hollmann et al., 2022; 2025) are transformer-based TFMs trained on massive synthetic priors. At inference time, the observed dataset of BO evaluations is fed as the context input, which acts as the optimization history. Each new sample is appended to this context, and a single forward pass produces updated predictive means and variances. Thus, although PFNs have frozen parameters, their predictions adapt dynamically to the growing context, mimicking Bayesian updating without retraining (Müller et al., 2023; Rakotoarison et al., 2024).

This approach eliminates iterative kernel re-fitting required by GPs, yielding 10–100× speedups in various BO applications (Müller et al., 2023; Rakotoarison et al., 2024; Yu et al., 2025). However, PFNs cannot explicitly tune kernels or priors, which limits their ability to exploit low-rank structures when dimensionality grows. Moreover, recent analyses reveal that while transformer-based PFNs exhibit vanishing variance with larger contexts, their bias persists unless explicit locality is enforced, resulting in degraded accuracy in high-dimensional regimes (Nagler, 2023). Although TabPFN v2 extends the model's capability to regression tasks with up to 500-dimensional inputs, its predictive performance still deteriorates without additional structural guidance (Ye et al., 2025; Reuter et al., 2025). These limitations highlight the necessity of incorporating additional guidance to sustain the effectiveness of TabPFN v2 in high-dimensional BO.

## 2.3 Discovering Embedded Subspaces: Classical and Deep Learning Perspectives

Since PFNs are frozen models, discovering intrinsic low-dimensional subspaces is the most viable high-dimensional BO strategy requiring no fine-tuning of the foundation model. Classical applied mathematics offers a principled method that we can leverage here. Active subspaces (Constantine et al., 2014) use gradient covariance to identify influential directions, while likelihood-informed subspaces (Cui et al., 2014) detect posterior-sensitive directions. Spectral approaches such as Laplacian eigenmaps (Belkin & Niyogi, 2001) learn nonlinear embeddings. Recent advances provide certified guarantees of gradient subspace recovery via $\phi$-Sobolev inequalities (Zahm et al., 2022; Li et al., 2024; 2025). In contrast, deep learning methods typically learn mappings into latent manifolds, e.g., variational autoencoders designed for BO (Tripp et al., 2020) or intrinsic-dimension analyses of neural representations (Li et al., 2018; Ansuini et al., 2019). These approaches require training additional models, which conflicts with the TFM paradigm of fast inference without retraining.

The literature suggests a potential synthesis: pair the inference-time efficiency of TFMs with structure discovery to address high-dimensional optimization. This leads to our central contribution: Gradient-Informed Bayesian Optimization using TabPFN (GIT-BO), which extracts predictive-mean gradients from TabPFN v2 to estimate a gradient-informed active subspace, then performs acquisition-driven search within that subspace. This design (i) avoids GP retraining and heavy hyperparameter tuning, and (ii) supplies the locality and structure that TFMs lack in high-dimensional—thereby targeting the exact failure modes surfaced above.

## 3 THE GIT-BO ALGORITHM

GIT-BO consists of four main components: the surrogate model (TabPFN v2), the gradient-based subspace identification, an upper confidence bound acquisition function, and a method that combines these for high-dimensional optimization.

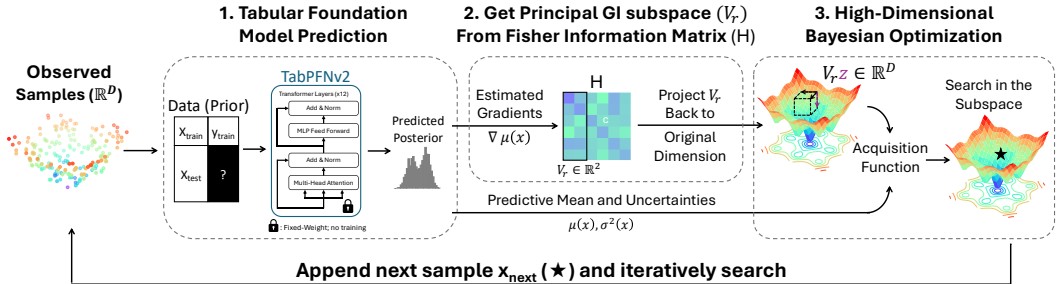

Figure 1: GIT-BO algorithm overview. The method operates in five stages: (1) Initial observed samples are collected in the high-dimensional space $\mathbb{R}^D$; (2) TabPFN v2, a fixed-weight tabular foundation model, generates predictions of the objective space at inference time using in-context learning; (3) The gradient from TabPFN's forward pass ($\nabla\mu(x)$) is used to identify a low-dimensional gradient-informed (GI) subspace. The predicted mean and variance are used for acquisition value calculations $\mu(x), \sigma^2(x)$; (4) The next sample point ($x_{\text{next}}$) is selected from GI subspace's projection back to the high-dimensional space ($V_r z$) with the highest acquisition value; (5) Appended $x_{\text{next}}$ to the "context" observed dataset for iterative search until stopping criteria is met.

### 3.1 SURROGATE MODELING WITH TABPFN

We use TabPFN v2, a 500-dimensional TFM from Hollmann et al. (2025), as the surrogate model for our Bayesian optimization framework. TabPFN leverages in-context learning to provide a dynamic predictive posterior distribution conditioned on the observed dataset $\mathcal{D}_{\text{obs}} = \{(x_i, y_i)\}_{i=1}^n$, which expands iteratively during optimization. At each iteration, we random sample (from Sobol sequence) a huge discrete set of candidate points $X_{\text{cand}} = \{x_j\}_{j=1}^m$ (where $(10k-n) \geq m \gg n$, as TabPFN can take at most 10k samples) from the search domain $\mathcal{X} \subset \mathbb{R}^D$ to approximate the continuous search space. TabPFN ($q_\theta$) processes both the context set ($\mathcal{D}_{\text{obs}}$) and candidate set ($X_{\text{cand}}$) simultaneously, generating predictive means $\mu_m(x)$ and variances $\sigma_m^2(x)$ for the search space formed by all candidates in a single forward pass ($\mu_m(x), \sigma_m^2(x) \sim q_\theta(Y_{\text{cand}}|X_{\text{cand}}, \mathcal{D}_{\text{obs}})$). This efficient, adaptive inference step enables rapid identification of promising regions in high-dimensional optimization problems without surrogate retraining.

### 3.2 GRADIENT-INFORMED ACTIVE SUBSPACE IDENTIFICATION AND SAMPLING

To identify an active subspace for efficient exploration, we leverage gradient information obtained from TabPFN's predictive mean, $\nabla_x \mu_m(x)$, computed through a single-step backpropagation. Using the theory from gradient active subspace methods for dimension reduction in nonlinear Bayesian inverse problems (Zahm et al., 2022; Li et al., 2025; 2024; Ly et al., 2017), we approximate the Fisher information matrix as $H = \mathbb{E}_\mu[\nabla_x \mu_m(x)\nabla_x \mu_m(x)^\top]$, which approximates covariance of the posterior distribution. With this approximated covariance matrix, the algorithm selects the top $r$ eigenvectors of $H$ as the gradient-informed active subspace (GI-subspace) $V_r \in \mathbb{R}^r$. For all the results we presented in Section 5 and Appendix D, we selected a fixed $r = 10$ for our experiment. Ablation studies on the effect of GI-subspace on BO performance and the selection of $r$ are presented in Appendix B.2.

Next, we then uniformly sample $m$ candidate points for exploration from the low-dimensional ($r$-dimensional) hypercube, $z \sim \text{U}([-1, 1]^r)$, and mapping these back to the original high-dimensional space via:

$$X_{\text{GI}} = x_{\text{ref}} + V_r z \ ,$$

the $m$ candidates are centered around the centroid of observed data, $x_{\text{ref}} = \bar{x}_{\text{obs}}$, which guides the search towards promising regions discovered so far, while the acquisition function later promotes exploitation. These generated candidates, $X_{\text{GI}}$, are then evaluated using the acquisition function to select the next point to sample. The theoretical detail and experimental results for the GI subspace are in Appendix A and C.

## 3.3 ACQUISITION FUNCTION

We adopt the Upper Confidence Bound (UCB) as our acquisition function, as heuristics BO and PFN-based BO both use in previous studies (Srinivas et al., 2010; Xu et al., 2025; Müller et al., 2023). UCB selects points by maximizing $\alpha_{\text{UCB}} = \mu(x) + \beta\sigma(x)$, where $\mu(x)$ denotes the surrogate's predictive mean, $\sigma(x)$ is surrogate's predictive standard deviation, and $\beta$ represents the exploration level. In our GIT-BO algorithm, $\mu(x)$ and $\sigma(x)$ are the TabPFN predictive mean and standard deviation given data $\mathcal{D}_{\text{obs}}$, and the $\beta$ is set to 2.33. Further details of $\beta$ ablation are in Appendix B.3 with theoretical analysis in Appendix A.

Putting everything together, Figure 1 and Algorithm 1 outline the GIT-BO procedure combining TabPFN with gradient-informed subspace search. The technical implementation details of GIT-BO are stated in the Appendix G.

---

**Algorithm 1** Gradient-Informed Bayesian Optimization using TabPFN (GIT-BO)

---

**Require:** objective $f$, domain $\mathcal{X} \subset \mathbb{R}^D$, initial sample size $n_0$, iteration budget $I$, subspace dimension $r$, $\alpha$ acquisition function
1: Draw $n_0$ LHS points $x_i$ and set $y_i = f(x_i)$; $D_n \leftarrow \{(x_i, y_i)\}_{i=1}^{n_0}$
2: **for** $i = 1$ to $I$ **do**
3:     $\mu_m$ and $\sigma_m^2 \leftarrow$ Fit TabPFN on $D_n$, and predict $X_{\text{cand}}$ randomly sampled from Sobol
4:     Calculate backprop gradient $\nabla_x\mu_m(x)$ from TabPFN's in-context learning of $D_n$
5:     Form Fisher information (diagnostic) matrix $H = \mathbb{E}_\mu[\nabla_x\mu_m(x)\nabla_x\mu_m(x)^\top]$
6:     $V_r \leftarrow$ top-$r$ eigenvectors of $H$
7:     $X_{\text{GI}} \leftarrow x_{\text{ref}} + V_r z$, with $z$ uniform sampled from the low-dim hypercube $z \sim \text{U}([-1, 1]^r)$
8:     $x_{\text{next}} \leftarrow \arg\max_j \alpha(X_{\text{GI}})$
9:     Evaluate $y_{\text{next}} = f(x_{\text{next}})$ and append the query point data $D_n \leftarrow D_n \cup \{(x_{\text{next}}, y_{\text{next}})\}$
10: **end for**
11: **return** $x^\star = \arg\max_{(x,y) \in D} y$

---

## 4 EXPERIMENT

This section outlines our empirical approach to evaluating and comparing different high-dimensional Bayesian optimization algorithms, highlighting the assessment of different algorithms' performance across a large number of complex synthetic and unique engineering benchmarks. To conduct a fair, comprehensive comparison, we benchmark GIT-BO against four other algorithms from the state-of-the-art BO library, BoTorch (Balandat et al., 2020), on 60 problems, and conduct a statistical ranking evaluation over experiment trials.

## 4.1 BENCHMARK ALGORITHMS

We benchmark GIT-BO against random search (Bergstra & Bengio, 2012) and four high-dimensional BO methods, including SAASBO (Eriksson & Jankowiak, 2021), TURBO (Eriksson et al., 2019), Vanilla BO for high-dimensional (Hvarfner et al., 2024), and BAxUS (Papenmeier et al., 2022) from the state-of-the-art (SOTA) PyTorch-based BO library BoTorch (Balandat et al., 2020). To ensure a fair comparison with our GPU-accelerated GIT-BO framework, we deliberately selected only algorithms that can be executed efficiently on GPUs, as runtime scalability is a central evaluation criterion. All methods were run on identical compute resources (one node with the same CPU and GPU specifications), and additional implementation details are provided in Appendix E.

## 4.2 TEST PROBLEMS

This study incorporates a diverse set of high-dimensional optimization problems, including 9 synthetic problems and 11 real-world benchmarks. Synthetic and scalable problems include: Ackley, Rosenbrock, Dixon-Price, Levy, Powell, Griewank, Rastrigin, Styblin-Tang, and Michalewicz. We note that this set of synthetic functions is taken from BoTorch (Balandat et al., 2020) with their default setting, and therefore all the baseline algorithms from BoTorch have been tested on this set of synthetic functions.

The rest of the application problems are collected from previous optimization studies and conference benchmarks: the power system optimization problems from CEC2020 (Kumar et al., 2020), Rover (Wang et al., 2018), MOPTA08 car problem (Jones, 2008), two Mazda car problems (Kohira et al., 2018), and Walker problem from MuJoCo (Todorov et al., 2012). As this study focuses on the high-dimensional characteristic of the problem, we make all our benchmark problems single and unconstrained for testing. Therefore, we have applied penalty transforms to all real-world problems with constraints and performed average weighting to the two multi-objective Mazda problems. Among the 20 benchmarks, 10 (Synthetic + Rover) are scalable problems. To evaluate the algorithms' performance with respect to dimensionality, we solve the scalable problems for $D = \{100, 200, 300, 400, 500\}$. Therefore, we have experimented with a total of $5 \times 10 + 10 = 60$ different variants of the benchmark problems. Details of benchmark selections and their implementation details are listed in Appendix F.

## 4.3 METHODS FOR ALGORITHM EXPERIMENT

The algorithm evaluation aims to thoroughly compare GIT-BO to current SOTA Bayesian optimization techniques. This study focuses on maximizing the objective function for the given test problems. For each test problem, our experiment consists of 20 independent trials, each utilizing a distinct random seed. To ensure fair comparison, we initialize each algorithm with an identical set of 200 samples, generated through Latin Hypercube Sampling with consistent random seeds across all trials. During each iteration, each algorithm selects one sample to evaluate next.

To execute this extensive benchmarking process, we utilized a distributed server infrastructure featuring Intel Xeon Platinum 8480+ CPUs and NVIDIA H100 GPUs. For all algorithms, each individual experiments (run) were conducted with the same amount of compute allocated: a single H100 GPU node with 24 CPU cores and 224GB RAM.

## 4.4 EVALUATION METRICS

**Optimization Fixed-budget Convergence Analysis** Fixed-budget evaluation is a standard technique for comparing the efficiency of optimization algorithms by allocating a predetermined amount of computational resources for their execution (Hansen et al., 2022). In our study, we adopt a fixed-iteration budget, running all algorithms (GIT-BO, SAASBO, TurBO, Vanilla BO, and BAxUS) for 500 iterations. We report performance using plots of average regret versus the number of function evaluations (iterations), which illustrate the convergence behavior of each algorithm.

In addition, we measure the wall-clock runtime of each algorithm over the same 500 iterations. To capture efficiency in terms of computational cost, we report plots of average regret versus elapsed runtime (in seconds).

**Statistical Ranking** To comprehensively compare and evaluate the performance of the Bayesian optimization algorithms, statistical ranking techniques are employed instead of direct performance measurements of the optimization outcome. In this study, we define the optimization performance result as the median of the optimal found across the 20 optimization trials of each algorithm. By statistically ranking the results, we were able to standardize the comparisons across different problems, since various optimization challenges can produce objective values of vastly different magnitudes. Furthermore, using this ranking allowed us to reduce the distorting effects of unusual or extreme data points that might influence our evaluation.

We conduct our statistical analysis using the Friedman and Wilcoxon signed-rank tests, complemented by Holm's alpha correction. These non-parametric approaches excel at processing bench-

marking result data without assuming specific distributions, which is critical for handling optimization results with outliers. These statistical methods effectively handle the dependencies in our setup, where we used the same initial samples and seeds to test all algorithms. The Wilcoxon signed-rank test addresses paired comparisons between algorithms, while the Friedman test manages problem-specific grouping effects. For multiple algorithm comparisons, we used Holm's alpha correction to control error rates (Wilcoxon, 1945; Holm, 1979).

# 5 RESULTS

**Overall Statistical Ranking and Algorithm Runtime Tradeoffs**   Across all benchmark variants, Figure 2 (a) shows that GIT-BO achieves the best overall statistical performance rank (1.92) across 60 problems, consistently outperforming competing baselines in terms of final solution quality after 500 iterations. In terms of computational cost, Figure 2 (b) demonstrates that GIT-BO remains runtime-competitive despite its stronger optimization performance. To provide further insight, Figures 2 (c) and (d) decompose performance by problem class: BAxUS achieves the best ranking on synthetic benchmarks, whereas GIT-BO dominates on real-world engineering tasks. This contrast underscores that methods excelling on or even finetuning toward synthetic tests may not generalize to practical applications.

The joint performance–runtime tradeoff is visualized in Figure 3 (a), where GIT-BO and TurBO both lie on the Pareto frontier: GIT-BO attains superior optimization quality, while TurBO provides a speed advantage. Finally, Figure 3 (b) tracks the evolution of average algorithm rank over iterations, showing that GIT-BO rapidly rises to the top within the first 50 iterations and maintains its lead thereafter. Together, these results highlight GIT-BO as the most balanced method, achieving state-of-the-art performance while retaining favorable computational efficiency.

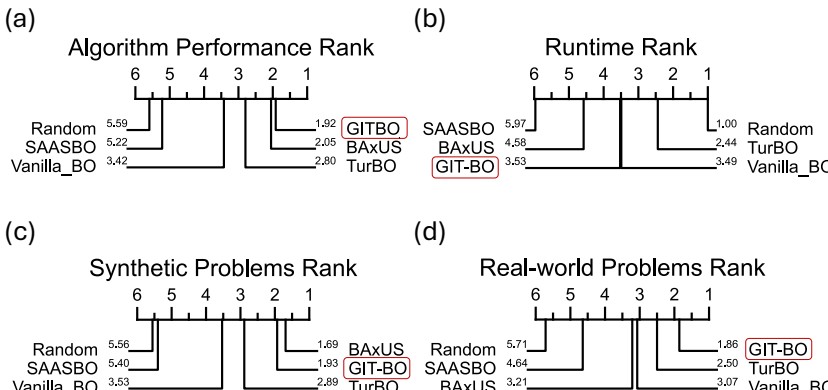

Figure 2: Statistical ranking across benchmark problems. (a) Overall algorithm optimization performance of all 60 problems (synthetic + real-world) ranking based on final solution quality at iteration 500. (b) Algorithm runtime ranking of all 60 problems (synthetic + real-world) based on the time it takes for 500 iterations of optimization. (c) and (d) Optimization performance ranking on only synthetic and only real-world benchmark subsets, respectively.

**Convergence Performance**   Figure 4 summarizes the convergence plots across a representative set of 15 synthetic and engineering benchmarks in iterations, and Figure 5 plots the average regret against the algorithm's elapsed runtime. Due to page number limitations, the convergence plots for all sixty benchmark problems are reported in Appendix D. For the Ackley function (100–500D), we observe that GIT-BO starts in the second performance tier but steadily improves relative to competing methods as dimensionality increases. Unlike GP-based approaches such as TurBO, whose performance deteriorates with higher $D$, GIT-BO maintains stable convergence rates, suggesting that TabPFN's universal modeling capacity generalizes robustly even in extreme dimensions.

Across the broader set of synthetic problems, GIT-BO achieves top-ranked regret curves in most cases, including Rosenbrock (200D), Dixon-Price (400D), and Rastrigin (500D). However, its failure on Styblinski–Tang highlights the distributional limits of the TabPFN pre-training regime, an

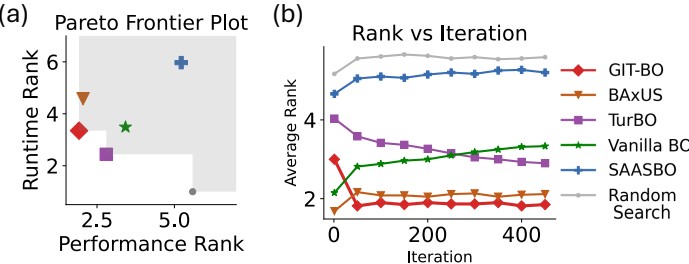

Figure 3: (a) Pareto frontier plot of runtime rank vs. performance rank (lower is better) over 60 benchmark problems. (b) Evolution of average algorithm rank over iterations, showing that GIT-BO converges rapidly to the top within 50 iterations and sustains its lead.

example where GP-based surrogates still dominate. On the engineering side, GIT-BO again demonstrates strong performance, consistently outperforming baselines on power system tasks and automotive design benchmarks, while struggling with the Rover problem.

**Convergence performance when considering runtime** When runtime is taken into account in Figure 5, the trade-off becomes even more pronounced. Methods such as BAxUS can match or occasionally surpass GIT-BO in final regret, but only after an additional hour of wall-clock time. In contrast, GIT-BO reaches competitive or superior regret levels within minutes, providing a decisive advantage in time-critical engineering settings. Taken together, these iteration- and runtime-based analyses establish GIT-BO as both the most efficient and broadly effective algorithm among current high-dimensional BO methods.

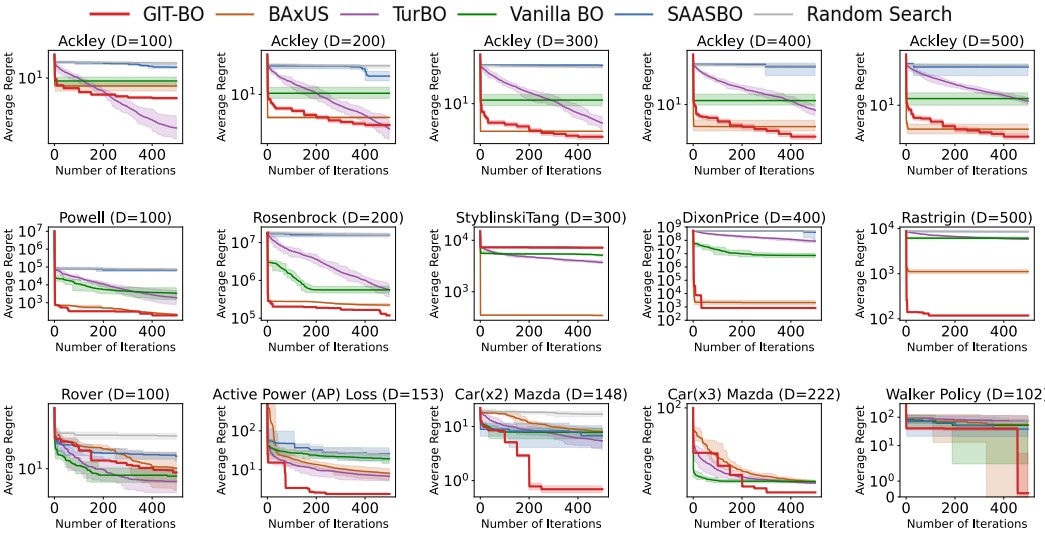

Figure 4: Average regrets vs. iteration convergence on a subset of 15 benchmarks (10 synthetic & 5 real-world) comparing our method against SOTA high-dimensional BO algorithms. The solid line represents the median best function value achieved over 20 trials, with shaded regions indicating the 95% confidence interval. Full statistical tests and per-problem plots for all 60 problems are provided in the Appendix D.

# 6 DISCUSSION

Our experiments highlight several strengths and limitations of GIT-BO in high-dimensional Bayesian optimization. GIT-BO consistently lies on the Pareto frontier of performance versus run-

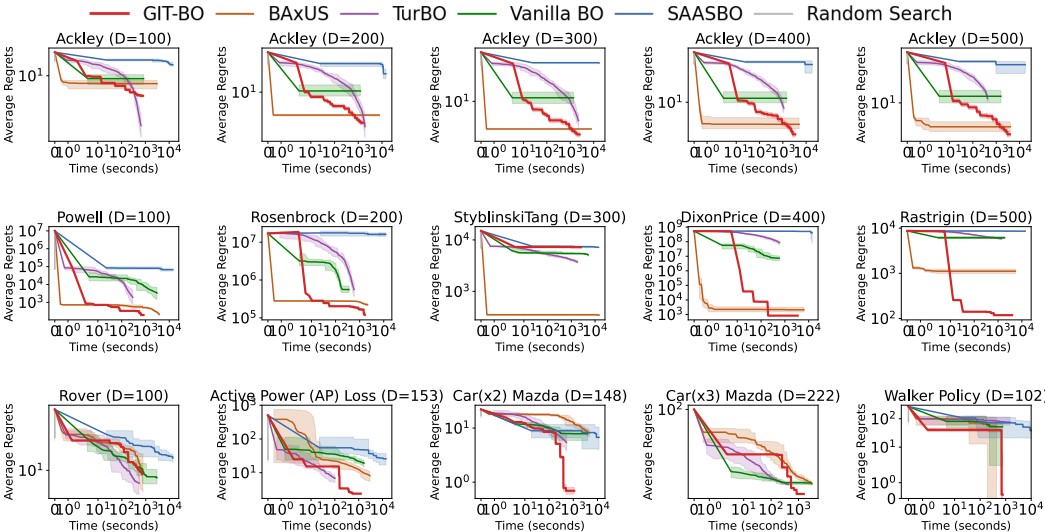

Figure 5: Average regrets (log-scaled) vs. algorithm runtime (log-scaled seconds) ("time taken for running 500 iterations") convergence on a subset of 15 benchmarks (10 synthetic & 5 real-world) comparing our method against SOTA high-dimensional BO algorithms. The solid line represents the median best function value achieved over 20 trials, with shaded regions indicating the 95% confidence interval. Full statistical tests and per-problem plots for all 60 problems are provided in the Appendix D.

time: while BAxUS and Vanilla BO can occasionally match final regret, they require orders of magnitude more wall-clock time, whereas GIT-BO reaches near-optimal solutions within minutes. At the same time, TurBO emerges as a compelling alternative when runtime alone is the dominant criterion, underscoring the practical trade-off between speed and accuracy. We also observe plateauing convergence in both GIT-BO and BAxUS, reflecting the known bias plateau of TabPFN predictors as sample sizes grow (Nagler, 2023) and pointing to broader challenges for probabilistic surrogates. Although GIT-BO excels on most synthetic and engineering tasks, its failures on Rover and Styblinski–Tang reinforce the "no free lunch" theorem (Wolpert & Macready, 1997). Finally, practical limits persist: TabPFN requires large GPU memory, enforces a 500D cap, and demands user-specified subspace thresholds. Even without retraining, its inference is slower than fitting a simple GP in TurBO or Vanilla BO. These findings suggest two directions for future work: scaling TFMs with memory-efficient architectures for faster inference, and designing benchmark suites that capture the heterogeneity of real-world tasks beyond synthetic testbeds.

# 7 CONCLUSION

We presented GIT-BO, a Gradient-Informed Bayesian Optimization framework that integrates TabPFN v2 with adaptive subspace discovery to tackle high-dimensional black-box problems. Across sixty benchmark variants, including scalable synthetic functions and challenging engineering tasks, GIT-BO consistently achieves state-of-the-art performance while maintaining a favorable runtime profile, often reaching near-optimal solutions in minutes. By leveraging foundation model inference and gradient-informed exploration, GIT-BO eliminates costly surrogate retraining and scales effectively up to 500 dimensions. At the same time, limitations remain: performance plateaus on certain tasks, GPU memory requirements of TabPFN, and the need for user-defined subspace thresholds. Looking forward, future work should pursue more memory-efficient TFM architectures, automated strategies for subspace selection, and broader benchmark suites that bridge synthetic testbeds and real-world engineering problems. Extending GIT-BO to constrained, mixed-variable, and multi-objective optimization also represents a promising avenue for further impact.

REPRODUCIBILITY STATEMENT

We are committed to ensuring reproducibility of all results. Upon acceptance, we will release the full source code, including the implementation of GIT-BO, data preprocessing scripts, benchmark configurations, and experiment pipelines. All experiments will be accompanied by fixed random seeds, hardware specifications, and detailed instructions to reproduce the results in the paper.

ETHICS STATEMENT

This work does not raise any ethical concerns.

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

TABLE OF CONTENTS FOR APPENDICES

# A  THEORETICAL ANALYSIS

In this section, we establish the theoretical foundations for GIT-BO by developing confidence bounds and regret guarantees. Our analysis builds upon the framework of Srinivas et al. (2010) for GP-UCB while accounting for the unique properties of TabPFN as a surrogate model and our gradient-informed subspace identification.

## A.1  PRELIMINARIES

**Problem Setup.**  We consider the optimization problem:

$$\boldsymbol{x}^* \in \arg\max_{\boldsymbol{x} \in \mathcal{X}} f(\boldsymbol{x})$$

where $\mathcal{X} = [0,1]^D$ is a compact domain and $f : \mathcal{X} \to \mathbb{R}$ is an unknown objective function. At each optimization iteration $t$, we observe $y_t = f(\boldsymbol{x}_t) + \epsilon_t$ where $\epsilon_t$ is $\sigma$-sub-Gaussian noise.

**TabPFN Surrogate Properties.**  Let $q_{\boldsymbol{\theta}}(y|\boldsymbol{x}, D_t)$ denote the TabPFN's posterior predictive distribution at point $\boldsymbol{x}$ given observed data $D_t = \{(\boldsymbol{x}_i, y_i)\}_{i=1}^t$. We denote the predictive mean and variance as:

$$\mu_t(\boldsymbol{x}) = \mathbb{E}_{q_{\boldsymbol{\theta}}}[y|\boldsymbol{x}, D_t], \quad \sigma_t^2(\boldsymbol{x}) = \mathrm{Var}_{q_{\boldsymbol{\theta}}}[y|\boldsymbol{x}, D_t]$$

**Reference GP class & Information Gain.**  For a kernel $k$ and noise $\sigma^2$, define the (maximal) information gain

$$\gamma_T = \max_{A:|A|=T} I(y_A; f_A) = \max_A \frac{1}{2} \log\det(I + \sigma^{-2} K_A). \tag{1}$$

In GP-UCB, cumulative regret admits the canonical bound $R_T = O(\sqrt{T\beta_T\gamma_T})$, with $\beta_T$ a confidence parameter depending on $\delta$ and the RKHS norm $\|f\|_k$ (Srinivas et al., 2010).

## A.2  ASSUMPTIONS

**Assumption 1** (TabPFN Approximation Quality). Based on the empirical results in Müller et al. (2022) and the statistical analysis from Nagler (2023) showing that TabPFN can approximate GP posteriors with high fidelity, there exists a constant $C_{\mathrm{approx}} > 0$ such that for any dataset $\mathbb{D}_t$ and query point $\boldsymbol{x} \in \mathbb{X}$:

$$\left| \mu_t(\boldsymbol{x}) - \mu_t^{GP}(\boldsymbol{x}) \right| \leq C_{\mathrm{approx}} \epsilon_{\mathrm{approx}}(t)$$

$$\left| \sigma_t^2(\boldsymbol{x}) - (\sigma_t^{GP}(\boldsymbol{x}))^2 \right| \leq C_{\mathrm{approx}} \epsilon_{\mathrm{approx}}(t)$$

where $\mu_t^{GP}(\boldsymbol{x})$ and $\sigma_t^{GP}(\boldsymbol{x})$ are the corresponding GP posterior mean and standard deviation, and $\epsilon_{\mathrm{approx}}(t) \to 0$ as the TabPFN training data size increases.

**Assumption 2** (Bounded Function Complexity). The true function $f$ has bounded RKHS norm: $\|f\|_k \leq B$ for some reproducing kernel $k$ and constant $B > 0$.

**Assumption 3** (Gradient-Informed Subspace). This assumption is based on Li et al. (2025; 2024). Let $\mu$ be a reference measure (e.g., standard Gaussian) and define the diagnostic/Fisher matrix

$$H = \mathbb{E}_\pi \left[ \nabla \log \ell(X) \nabla \log \ell(X)^\top \right], \quad \text{with } d\pi(x) \propto \ell(x) \, d\mu(x).$$

Let $V_r \in \mathbb{R}^{D \times r}$ contain the top-$r$ eigenvectors of $H$. Then the best $r$-dimensional ridge approximation $\tilde{\pi}_r$ to $\pi$ enjoys a certified error

$$D_\alpha(\pi \| \tilde{\pi}_r) \leq \mathcal{J}_\alpha \left( C_\alpha(\mu) \sum_{k=r+1}^{D} \lambda_k(H) \right),$$

for all $\alpha \in (0, 1]$, where $\lambda_k(H)$ are the eigenvalues of $H$ in descending order and $C_\alpha(\mu)$ depends only on $\mu$. Thus choosing $V_r$ by Fisher-eigenvectors minimizes a tight majorant of the divergence,

with sharper (dimensional) certificates available for $\alpha = 1$ (KL). We use this to quantify subspace truncation error.

The certificate above follows from $\varphi$-Sobolev / logarithmic-Sobolev bounds that (i) deliver the same $V_r$ for KL and Hellinger and (ii) upper-bound the divergence by the tail trace $\sum_{k>r} \lambda_k(H)$ (Li et al., 2024). Dimensional LSI further sharpens the KL majorant and yields matching minorants at the minimizer (Li et al., 2025).

### A.3 CONFIDENCE BOUNDS FOR TABPFN-BASED SURROGATES

Define:

$$\beta_t = 2B^2 + 2\log\left(\frac{\pi^2 t^2}{3\delta}\right) + 2C_{\text{approx}}^2 \epsilon_{\text{approx}}^2(t)$$

**Lemma 1** (TabPFN-UCB Confidence Bounds). With probability at least $1 - \delta$, for all $t \geq 1$ and $\boldsymbol{x} \in \mathcal{X}$:

$$|f(\boldsymbol{x}) - \mu_t(\boldsymbol{x})| \leq \sqrt{\beta_t}\sigma_t(\boldsymbol{x})$$

This follows the martingale concentration approach of Srinivas et al. (2010) but includes an additional approximation error term $C_{\text{approx}}\epsilon_{\text{approx}}(t)$ to account for the difference between TabPFN and the ideal GP posterior. The bounded RKHS norm assumption ensures the function lies in a well-defined function class, while the approximation quality assumption controls the deviation from GP-based confidence bounds.

### A.4 SUBSPACE INFORMATION GAIN ANALYSIS

To analyze GIT-BO's regret, we must characterize how much information can be gained about the objective function when optimization is restricted to the gradient-informed subspace.

**Definition 1** (Subspace Information Gain). For a subspace $\mathbb{S} \subset \mathbb{X}$ and set of points $A = \{\boldsymbol{x}_1, \ldots, \boldsymbol{x}_T\} \subset \mathbb{S}$, the subspace information gain is:

$$\gamma_{T,\mathbb{S}} := \max_{A \subset \mathbb{S}, |A|=T} I(y_A; f_A)$$

where $I(y_A; f_A) = \frac{1}{2}\log|\boldsymbol{I} + \sigma^{-2}K_A|$ is the mutual information between observations $y_A$ and function values $f_A$.

**Lemma 2** (Subspace Approximation Error). Under Assumption 3, the approximation error for restricting optimization to the gradient-informed subspace $V_r$ satisfies:

$$D_{KL}(\pi_{\text{full}}\|\pi_r) \leq \frac{1}{2}\sum_{k=r+1}^{d}\lambda_k(H)$$

where $\pi_{\text{full}}$ represents the target distribution in the full space and $\pi_r$ is its approximation in the subspace $V_r$.

**Lemma 3** (Subspace Information Gain Bound). Under Assumption 3, the information gain in the gradient-informed subspace $V_r$ satisfies:

$$\gamma_{T,V_r} \geq \alpha\gamma_{T,\text{full}} - C_{\text{sub}}T^{1/2}$$

where $\gamma_{T,\text{full}}$ is the information gain in the full space and $C_{\text{sub}}$ is a constant depending on the subspace construction quality. This follows from Assumption 3, which ensures that the Fisher eigenvectors $V_r$ minimize a Sobolev-type divergence between the full distribution and its subspace projection. The information gain in $V_r$ is therefore lower-bounded by the variation captured in the retained eigenvalues, linking subspace structure directly to the information-theoretic quantity.

A.5   ACQUISITION FUNCTION ANALYSIS

We adopt the Upper Confidence Bound (UCB) acquisition, a standard principle in Bayesian optimization (Srinivas et al., 2010). At each iteration $t$, given a predictive posterior with mean $\mu_t(x)$ and standard deviation $\sigma_t(x)$, UCB selects:

$$x_t = \arg\max_{x \in \mathcal{X}} \alpha_t(x), \quad \alpha_t(x) = \mu_t(x) + \beta_t \sigma_t(x),$$

where $\beta_t > 0$ balances exploration and exploitation.

We instantiate $\beta_t$ in two equivalent ways:

**Definition 1** (Sampling-UCB). Draw $S$ i.i.d. samples $\tilde{y}_t(x) \sim \mathcal{N}(\mu_t(x), \sigma_t^2(x))$ and set

$$\alpha_t(x) = \max_{i=1,\ldots,S} \tilde{y}_i(x).$$

By extreme-value theory, the corresponding exploration parameter satisfies

$$\beta_t \approx \Phi^{-1}\left(1 - \frac{1}{S}\right),$$

which asymptotically behaves as $\sqrt{2 \log S}$ with standard corrections (Srinivas et al., 2010).

**Definition 2** (Quantile-UCB fro (Müller et al., 2023)). For a one-sided Gaussian quantile $q \in (0, 1)$, set

$$\beta_t = \Phi^{-1}(q),$$

where $\Phi^{-1}$ is the standard normal inverse CDF. Then

$$\alpha_t(x) = \text{Quantile}_q \left[ \mathcal{N}(\mu_t(x), \sigma_t^2(x)) \right].$$

This corresponds to selecting the $q$-th posterior quantile, with higher $q$ producing more exploration. In code, this is parameterized by a "rest probability" $p_{\text{rest}}$, where $q = 1 - p_{\text{rest}}$.

**Lemma 4** (Equivalence). Quantile-UCB with quantile level $q = 1 - 1/S$ is asymptotically equivalent to Sampling-UCB with $S$ posterior draws. Both implement the same exploration policy, differing only in whether the quantile is computed analytically or via sampling.

**Remark A.5.**   In practice, we adopt the sampling formulation of UCB, which introduces mild stochasticity by drawing finite posterior samples. This choice yields trajectories that may vary more across runs, akin to the exploratory effect of UCB. By contrast, the quantile formulation produces a deterministic acquisition rule given the posterior, leading to more stable and less variable optimization behavior. We provide an ablation of both variants in Appendix B.3. Presenting the two side by side highlights their close equivalence while ensuring transparency in how exploration is controlled.

A.6   MAIN REGRET BOUNDS

We now establish our main theoretical result for GIT-BO's regret performance.

**Theorem 1** (GIT-BO Regret Bound). Under Assumptions 1-3, let $\delta \in (0, 1)$ and run GIT-BO with confidence parameter:

$$\beta_t = 2B^2 + \sqrt{2 \log S} + 2 \log\left(\frac{\pi^2 t^2}{3\delta}\right) + 2C_{\text{approx}}^2 \epsilon_{\text{approx}}^2(t)$$

Then with probability at least $1 - \delta$, the cumulative regret after $T$ iterations satisfies:

$$R_T \leq \sqrt{C_1 T \beta_T \gamma_{T,V_r}} + \sum_{t=1}^{T} (1 - \alpha) \sqrt{\beta_t \sigma_t^2(\boldsymbol{x}_t)} + T C_{\text{approx}} \epsilon_{\text{approx}}(T)$$

where $C_1 = 8/\log(1 + \sigma^{-2})$ and the second term accounts for subspace approximation error.

### A.7 INFORMATION GAIN BOUNDS FOR HIGH-DIMENSIONAL SUBSPACES

**Lemma 5** (Polynomial Information Gain). For common kernel functions (RBF, Matérn) restricted to an $r$-dimensional subspace where $r \ll D$, the information gain satisfies:

$$\gamma_{T,V_r} = O(r(\log T)^{r+1})$$

This represents a significant improvement over the full-dimensional case where $\gamma_{T,\text{full}} = O(D(\log T)^{D+1})$. This follows the spectral analysis of kernel functions in lower-dimensional spaces, adapting the techniques of Srinivas et al. (2010) to the subspace setting.

### A.8 CONVERGENCE RATE

Combining our results, we obtain the following convergence guarantee:

**Corollary 1** (Convergence Rate). Under the conditions of Theorem 1, if the TabPFN approximation error satisfies $\epsilon_{\text{approx}}(t) = O(t^{-\xi})$ for some $\xi > 1/2$, then:

$$\lim_{T \to \infty} \frac{R_T}{T} = 0$$

with convergence rate $R_T = O(\sqrt{rT(\log T)^{r+2}})$ when $r \ll D$.

This demonstrates that GIT-BO achieves sublinear regret with dimension-independent rates when the effective dimensionality $r$ is small, addressing the curse of dimensionality that plagues standard GP-based methods.

## B ABLATION STUDIES

### B.1 WHY UCB AND WHY GI SUBSPACE? — ABLATION STUDY ON DIFFERENT ACQUISITION FUNCTIONS AND THE EFFECT OF GI SUBSPACE FOR HIGH-DIMENSIONAL BO

We compared GIT-BO against vanilla TabPFN v2 without subspace identification, each paired with either EI or UCB. As shown in Figure 6, both EI and UCB with the vanilla model fail in high dimensions, converging slowly and to worse optima. Adding GI subspaces restores stable convergence. On engineering tasks, EI and UCB perform similarly, while on the multimodal Ackley function, UCB shows a slight edge. These results confirm that TabPFN v2 alone cannot capture key search directions at scale, and that GI subspace refinement is the critical ingredient behind GIT-BO's robustness

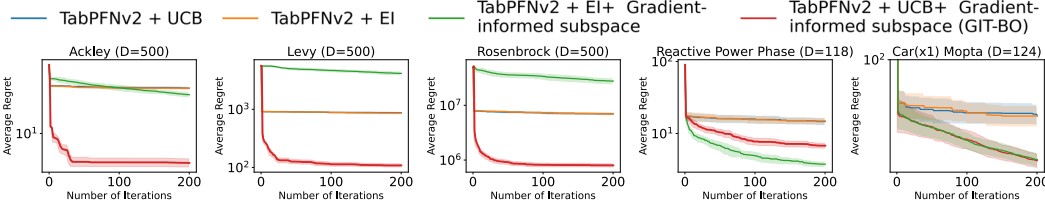

Figure 6: Ablation studies of vanilla TabPFN v2 BO with GIT-BO. GIT-BO, regardless of the acquisition functions, consistently outperforms other algorithms without GI subspace search across all tested problems (8.6 times better regrets), indicating vanilla TabPFN v2 is not an ideal candidate for performing high-dimensional BO without GI subspace search.

### B.2  WHY $r = 10$? — PARAMETER SWEEP ABLATION OF GI SUBSPACE'S PRINCIPAL DIMENSION $r$

To evaluate the sensitivity of GIT-BO to the dimensionality of the gradient-informed active subspace, we conducted a parameter sweep across both fixed subspace dimensions ($r = 5, 10, 15, 20, 40$) and variance-explained criteria ($92.5\%, 95\%, 97.5\%$) for a subset of four problems. The results, summarized in Figure 7 and Table 1, highlight two consistent trends. First, very high-dimensional subspaces (e.g., $r = 40$) exhibit clear performance degradation, indicating that overly broad subspaces dilute the effectiveness of the gradient-informed search direction. Second, low- to moderate-dimensional subspaces and variance-based selections generally perform better, though the best choice of $r$ varies across problem families. For example, $r = 5$ yields the top average rank among different $r$s, while variance-based selection at the $92.5\%$ and $95\%$ thresholds achieves the top overall results.

To ensure fairness and avoid additional hyperparameter tuning, we fixed $r = 10$ for all benchmarks reported in the main text. This choice provides a stable middle ground, neither overly restrictive nor excessively large, while still yielding competitive performance across diverse problem classes. Notably, adaptive variance-based selection strategies further improve performance on average, underscoring the potential benefit of problem-dependent tuning, but we leave such extensions for future work. Overall, these ablation results confirm that GIT-BO remains robust to the specific choice of $r$, with consistent advantages over GP-based baselines even under a fixed setting.

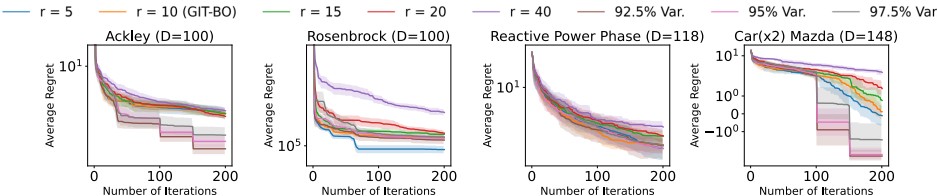

Figure 7: Performance of GIT-BO under different gradient-informed subspace dimensions ($r$) and variance-explained criteria. Median optimization regret across 20 trials is shown, with shaded regions denoting 95% confidence intervals. High-dimensional subspaces (e.g., $r = 40$) consistently degrade performance, while smaller fixed dimensions and adaptive variance-based selections achieve stronger results. We fix $r = 10$ across all benchmarks in the main text for fairness, as it provides a balanced and competitive setting without tuning.

Table 1: Average performance rank of GIT-BO across different fixed subspace dimensions $r$ and variance-based adaptive selections.

| Selection of $r$ | Average Rank |
| --- | --- |
| 92.5% Variance | 1.75 |
| 95% Variance | 2.25 |
| 97.5% Variance | 3.5 |
| $r = 5$ | 3.25 |
| $r = 10$ | 5.5 |
| $r = 15$ | 5.5 |
| $r = 20$ | 6.35 |
| $r = 40$ | 8.0 |

### B.3  WHY SAMPLING-BASED UCB — ABLATION STUDY ON DIFFERENT $\beta$ FACTOR OF UCB ACQUISITION FUNCTION

We compared two equivalent parameterizations of UCB: (1) quantile-UCB, which uses the analytic Gaussian quantile, and (2) sampling-UCB, which approximates it via the maximum over $S$ posterior draws. Both induce similar exploration levels for $\alpha = 1 - 1/S$, but differ in that sampling introduces mild stochasticity. Our ablation results in Figure 8 and Table 2 shows that moderate exploration ($\beta \approx 1.86 - 1.96$, i.e., quantile 95% - 97.5% or sampling with $S \approx 250$) achieves the best ranks. Larger $\beta$ values ($S = 512, 1024$) lead to over-exploration and degraded performance.

In the main body of the paper we pre-committed to a single, conservative default ($S = 512$) across all 60 tasks and 500 iterations per task. We did this deliberately for three reasons: 1. Fairness and reproducibility: Using one global setting avoids per-benchmark tuning (or hindsight "cherry picking") and makes results easy to reproduce and audit across a large suite. 2. Isolating the algorithmic contribution: We wanted to attribute gains to the proposed GI subspace + TabPFN framework rather than to problem-specific hyperparameter search. A fixed $\beta$ keeps the evaluation focused on the method, not tuning effort. 3. Practicality and compute parity: Sweeping $\beta$ across 60 problems $\times$ 500 iterations would multiply the already substantial compute; fixing a robust default is closer to how one would deploy the method under realistic constraints.

Despite this conservative choice (which the ablation shows is not the best), GIT-BO still outperformed all baselines in our main results. The ablation simply reveals additional headroom: modestly smaller $\beta$ values improve performance further. Designing an *automatic* $\beta$ adaptation (e.g., schedule or data-driven calibration) is promising future work, but is orthogonal to the core contribution and therefore left out of the main comparison.

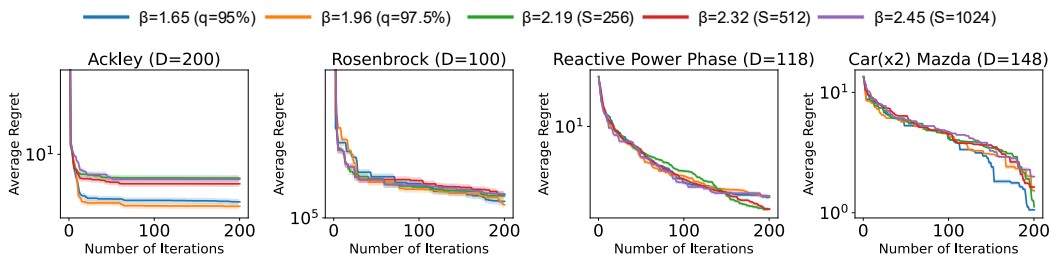

Figure 8: Ablation study of UCB with different exploration factors ($\beta$) using quantile- and sampling-based parameterizations. Moderate $\beta$ values (quantile 95–97.5% or sampling $S = 250$) yield the best performance, while larger $\beta$ (e.g., $S = 512, 1024$) leads to over-exploration and weaker results

Table 2: Average performance rank of GIT-BO across different $\beta$ from quantile-based and sampling-based UCB.

| Selection of $\beta$ | Average Rank |
|---|---|
| $\beta = 1.65$ ($q = 95\%$) | 2.0 |
| $\beta = 1.96$ ($q = 97.5\%$) | 2.25 |
| $\beta = 2.19$ ($S = 256$) | 2.75 |
| $\beta = 2.32$ ($S = 512$) | 3.25 |
| $\beta = 2.45$ ($S = 1024$) | 4.75 |

## B.4 WHY UNIFORM SAMPLING IN THE GRADIENT-INFORMED SUBSPACE? — ABLATION STUDIES OVER GI SUBSPACE SAMPLING

We conducted an ablation study to evaluate the impact of three different GI subspace sampling methods on GIT-BO's optimization performance: uniform (default), random, and Sobol sampling. Figure 9 shows the comparative convergence results.

Our findings indicate mixed results without a universally optimal sampling strategy. Uniform sampling generally provided stable and reliable convergence, while random sampling occasionally achieved better outcomes but with greater variance, similar as Sobol sampling. These observations highlight the potential for adaptive strategies in selecting GI subspace sampling methods based on problem-specific characteristics, representing an important area for future exploration.

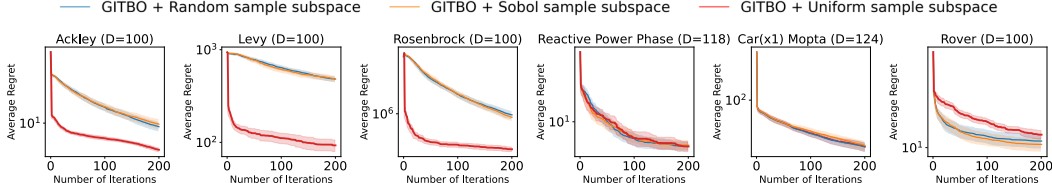

Figure 9: Comparative convergence of uniform, random, and Sobol sampling strategies within the GI subspace on selected benchmarks. Shaded regions represent 95% confidence intervals over 20 trials. Random and Sobol sampling can achieve similar or superior performance than uniform sampling GI subspace in engineering problems, while struggling at the synthetic tasks.

## C  EXPERIMENTAL ANALYSIS ON GRADIENT-INFORMED ACTIVE SUBSPACE IDENTIFICATION

A central question for gradient-informed (GI) subspace identification is whether it can reliably recover the intrinsic dimensionality of a problem when the objective is embedded in high dimensions. In principle, eigenvalue thresholds on gradient covariance spectra might fail—oscillating around spurious directions or overestimating dimensionality—unless the surrogate provides sufficiently smooth and informative gradients. We were therefore curious to test whether GIT-BO's GI subspace mechanism can autonomously identify the correct intrinsic dimension or not.

To probe this, we evaluate on Branin ($d=2$ embedded in 100D), Ackley ($d=3$ embedded in 100D), and Lévy ($d=3$ embedded in 200D). GIT-BO with TabPFN surrogates consistently auto-selects a subspace dimension $r$ (via a 95% variance threshold on the gradient covariance spectrum) that converges to the ground-truth $d$ after $\sim$50 iterations, while simultaneously reducing regret. These results suggest that TabPFN provides a smooth and informative gradient field that allows the GI subspace to identify the correct intrinsic structure of a problem, enabling efficient search in that space.

In contrast, when we replace TabPFN v2 with a basic GP surrogate, the GI subspace mechanism is far less effective. We implemented an untuned vanilla GP based on GPyTorch regression [1] with Expected Improvement (EI) [2]. To the best of our knowledge, BoTorch's GP models do not support BO with gradient observations, which was indicated in their GitHub issues too [3]. Therefore, we implemented GI subspace identification using this GP's gradients. The results show that GI subspace mechanism is far less effective: $r$ oscillates for longer periods, often fails to match the true $d$, and regret reduction stagnates. We hypothesize that this behavior arises from the following factors: (1) the 95% explained-variance threshold not being universally suitable for GPs; (2) in high dimensions, a plain GP with EI struggles because its hyperparameters are poorly estimated, leading to noisy or isotropic gradient fields; and (3) unlike TabPFN's amortized transformer prior, which yields globally informative gradients, standard GP kernels provide unreliable gradient estimates without stronger priors. These findings highlight that the performance of GI subspace identification is fundamentally tied to the quality of the surrogate's gradient field and motivate future work on gradient-accessible, high-dimensional GP models with stronger priors (e.g., SAASBO) or trust-region refinements (e.g., TuRBO).

## D  PERFORMANCE AND RUNTIME RESULTS ANALYSIS ON 60 BENCHMARKS

We report comprehensive optimization outcomes across all benchmark problems considered in the main paper. Figure 11 presents regret trajectories on all 60 benchmarks, and Figure 12 compares regret versus runtime.

---

[1] GPyTorch Tutorial (Batch Independent Multioutput GP)

[2] Expected Improvement code

[3] BoTorch GitHub Issue #1626

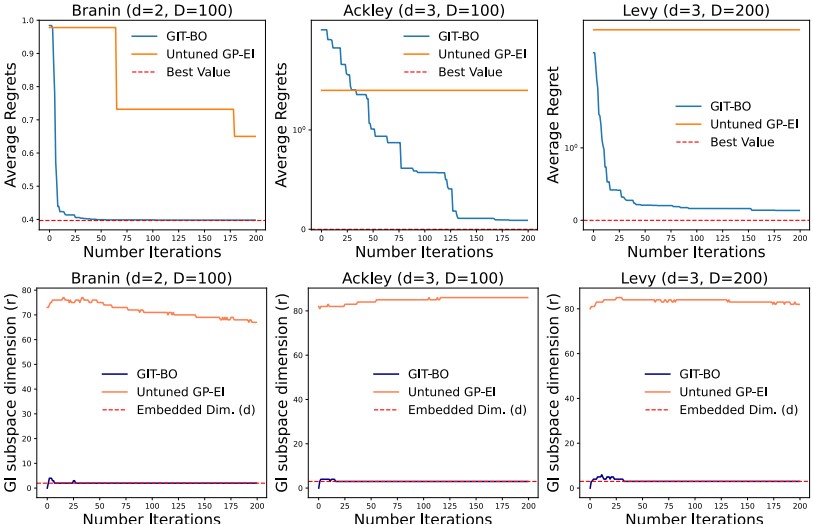

Figure 10: GI subspace behavior on high-dimensional embeddings. **Top:** Median (average) regret (20 trials, 95% CI). **Bottom:** Auto-selected subspace dimension $k$ under a 95% variance rule. TabPFN+GI converges to the correct intrinsic dimension ($k \to d$) with strong regret reduction, while GP(EI)+GI shows unstable $k$ and weaker optimization.

Overall, GIT-BO exhibits consistently strong performance across diverse high-dimensional problems, with clear advantages on most engineering benchmarks (with the exception of the Rover family). This highlights its ability to balance convergence speed and final solution quality relative to other state-of-the-art (SOTA) methods.

For synthetic problems, GIT-BO maintains robustness as dimensionality increases, whereas competing methods degrade more noticeably. Nevertheless, there are cases where GIT-BO underperforms across all $D$ (e.g., Styblinski–Tang and Michalewicz), consistent with the "No Free Lunch" theorem (Wolpert & Macready, 1997). We also observe plateauing in the convergence of both BAxUS and GIT-BO. For GIT-BO, this behavior is aligned with the known bias plateau of TabPFN predictors under increasing sample sizes (Nagler, 2023). The similar plateau in BAxUS suggests a broader phenomenon affecting probabilistic surrogates that merits further investigation.

On real-world engineering problems, GIT-BO ranks first overall, despite poor performance on the Rover tasks, again reinforcing "No Free Lunch." Interestingly, BAxUS, which dominates synthetic benchmarks, drops to fourth place on engineering problems. This discrepancy underscores the gap between synthetic and real-world benchmarks and motivates the need for more optimization benchmark design and evaluation.

## E  HIGH-DIMENSIONAL BENCHMARK ALGORITHMS IMPLEMENTATION DETAILS

We benchmark GIT-BO against four high-dimensional BO methods that GPU can also accelerate compute using PyTorch, including TURBO (Eriksson et al., 2019), Vanilla BO (Hvarfner et al., 2024), BAxUS (Papenmeier et al., 2022), and SAASBO (Eriksson & Jankowiak, 2021).

- TURBO: The implementation is taken from BoTorch's GitHub repository (Balandat et al., 2020) (link: `https://github.com/pytorch/botorch/blob/main/tutorials/turbo_1/turbo_1.ipynb`, license: MIT license, last accessed: Sep 21st, 2025)

- Vanilla BO: The implementation is taken from (Balandat et al., 2020) BoTorch version 13's GitHub repository (link: `https://github.com/pytorch/botorch/discussions/2451`, license: MIT license, last accessed: Sep 21stt, 2025)

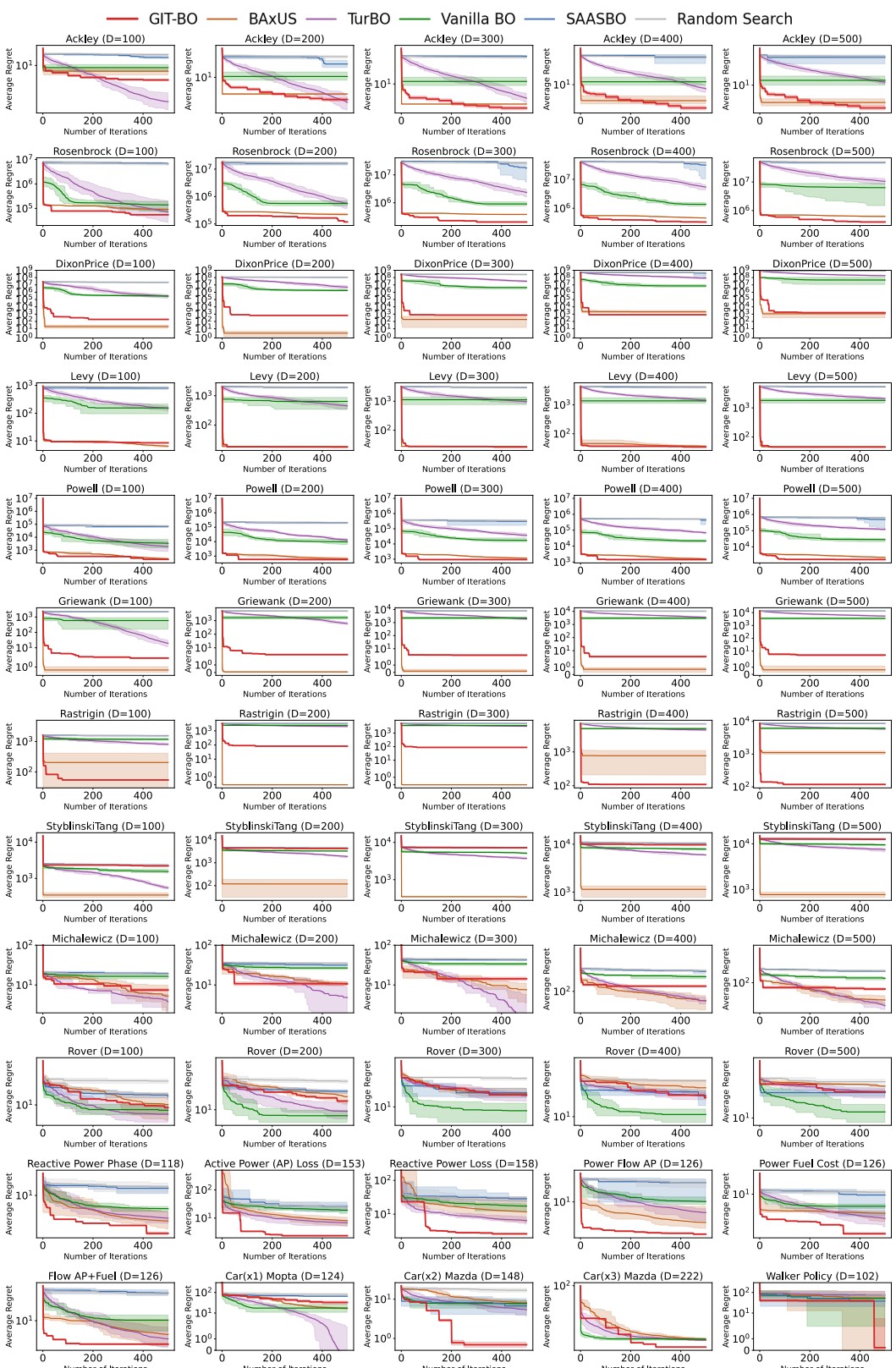

Figure 11: Average (median) regret vs. iterations (# function evaluations) with a budget of 500 iterations for all benchmarks. Average regrets are illustrated by solid lines, with shaded bands denoting 95% confidence intervals. The y-axis is log-scaled. GIT-BO finds the optimal value for 29 out of the total 60 problems.

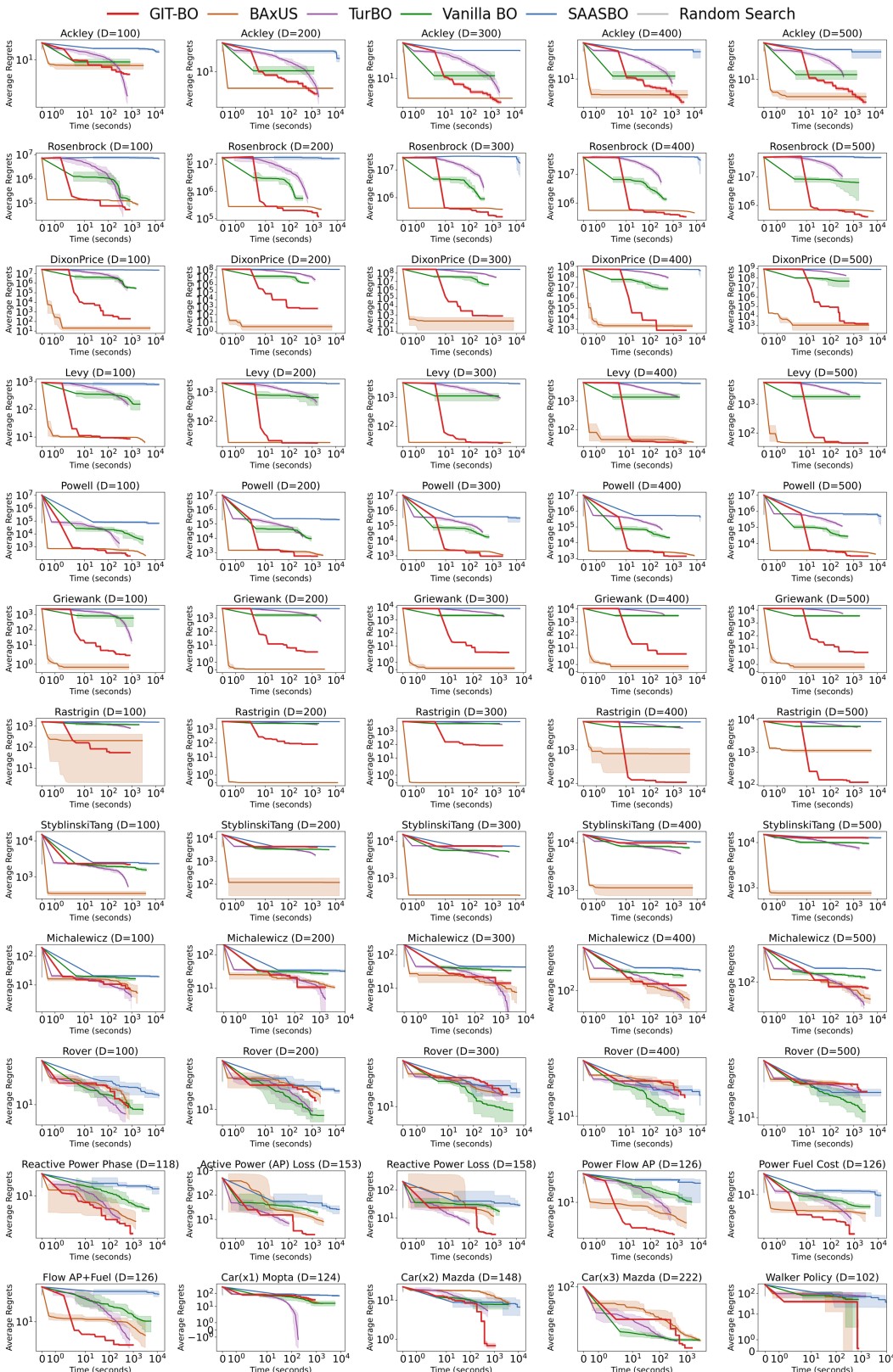

Figure 12: Average (median) regret vs. algorithm runtime (seconds) records of running 500 iterations for all benchmarks. Average regrets are illustrated by solid lines, with shaded bands denoting 95% confidence intervals. Both axes are log-scaled.

- BAxUS: The implementation is taken from BoTorch's GitHub repository (Balandat et al., 2020) (link: `https://github.com/pytorch/botorch/blob/main/tutorials/baxus/baxus.ipynb`, license: MIT license, last accessed: Sep 21st, 2025)

- SAASBO: We use the SAASBO-MAP version of the algorithm for comparison. The code is taken from Xu et al. (2025) (link: `https://github.com/XZT008/Standard-GP-is-all-you-need-for-HDBO/` commit b60e1c6, license: no license, last accessed: Sep 21st, 2025) where they implemented the MAP estimation of SAASBO based on the original paper (Eriksson & Jankowiak, 2021), with all the hyperparameter settings following precisely the same as the original paper. The reason for not using SAASBO-NUT is due to our computational resource limitations. We set a fixed maximum time of 10000 seconds for each trial to run, since running all benchmarking experiments takes roughly 72 million GPU compute seconds. Unfortunately, SAASBO-NUT can only run $\sim$310 iterations given this time budget, making it unfeasible for comparison with other algorithms that can finish running 500 iterations under 10000 seconds.

We use Botorch v0.12.0 for all algorithms mentioned above. The environment setups are detailed in the provided code zip file.

## F    BENCHMARK PROBLEMS IMPLEMENTATION DETAILS

The source and license details of our benchmark problems are provided in the following paragraphs. We restrict our evaluation to problems with well-maintained, publicly available code to ensure reproducibility and stability across our benchmark framework. Benchmarks that require complex or incompatible environment configurations are not included in the present study. Looking ahead, we advocate for a standardized collection of benchmarks with actively maintained codebases to facilitate broader adoption and more rigorous comparisons in future research. If this paper is accepted, we will release our Python benchmark library on PyPI alongside the publication.

**Synthetic Problems:**    The implementations for the nine synthetic functions are taken from Botorch (Balandat et al., 2020) (link: `https://github.com/pytorch/botorch/blob/main/botorch/test_functions/synthetic.py`, license: MIT license, last accessed: May 1st, 2025). The bounds of each problem are the default implementation in Botorch. Detailed equations for each problem can be found here: `https://www.sfu.ca/~ssurjano/optimization.html`.

**Power System Problems:**    We examine a subset of six problems, specifically those with design spaces exceeding 100 dimensions, from the CEC 2020 Real World Constrained Single Objective problems test suite (Kumar et al., 2020) (link: `https://github.com/P-N-Suganthan/2020-RW-Constrained-Optimisation`, license: no license, last accessed: May 1st, 2025). The code is initially in MATLAB, and we translate it into Python, running pytest to ensure the implementations are correct. While these problems incorporate equality constraints ($h_j(x)$), they are transformed into inequality constraints ($g_j(x)$) using the methodology outlined in the original paper (Kumar et al., 2020), as constraint handling is not the primary focus of this research. These transformed constraints are subsequently incorporated into the objective function $f(x)$ as penalty terms.

$$g_j(x) = |h_j(x)| - \epsilon \leq 0 \; , \; \epsilon = 10^{-4} \; , \; j = 1 \sim C$$

$$f_{penalty}(x) = f(x) + \rho \sum_{j=1}^{C} max(0, g_j(x))$$

We set a different $\rho$ penalty factor for each problem, respectively, to make the objective and constraint values have a similar effect on $f_{penalty}(x)$.

Table 3: Penalty Transform Factors of Benchmark Problems from CEC 2020$\rho$

| CEC's Problem Index | Our Naming | $\rho$ |
|---|---|---|
| 34 | Reactive Power Phase | 0.01 |
| 35 | Active Power (AP) Loss | 0.0002 |
| 36 | Reactive Power Loss | 0.001 |
| 37 | Power Flow AP | 0.04 |
| 38 | Power Fuel Cost | 0.02 |
| 39 | Power AP+Fuel | 0.04 |

**Rover:** The implementation is taken from Wang et al. (2018) (link: `https://github.com/zi-w/Ensemble-Bayesian-Optimization`, license: MIT license, last accessed: May 1st, 2025).

**Car(x1) Mopta:** The MOPTA08 is originally proposed by Jones (2008). The executable used in this study are taken from the paper Papenmeier et al. (2022)'s personal website (link: `https://leonard.papenmeier.io/2023/02/09/mopta08-executables.html`, license: no license, last accessed: May 1st, 2025). The MOPTA08 Car's penalty transformation follows the formation of Eriksson & Jankowiak (2021)'s supplementary material of a one-car car crash design problem.

**Car(x2) and Car(x3) Mazda Cars Benchmark Problems:** The implementation is taken from Kohira et al. (2018) (link: `https://ladse.eng.isas.jaxa.jp/benchmark/`, license: no license, last accessed: May 1st, 2025). The Mazda problem has two raw forms: a 4-objectives problem 148D optimizing a two-car car design problem (Car(x2)) and a 5-objectives 222D problem three-car car design problem (Car(x3)), and both of them have inequality constraints. For both problems, we equally weight each objective to form a single objective and perform a penalty transform:

$$f_{multiobj\_penalty}(x) = \frac{1}{N} \sum_{i=1}^{N} f(x) + \rho \sum_{j=1}^{C} max(0, g_j(x))$$

where $N$ is the number of objectives, $C$ is the number of inequality constraints, and we use $\rho = 10$ for both variants of Mazda problem.

**Walker Policy:** The problem is originally a locomotion task from MuJoCo (Multi-Joint dynamics with Contact) physics engine (Todorov et al., 2012) (Walker-2D), one of the most popular Reinforcement Learning (RL) benchmarks. The implementation of this RL policy search problem is directly taken from Wang et al. (2020) (link: `https://github.com/facebookresearch/LA-MCTS/tree/main/example/mujuco`, license: CC-BY-NC 4.0 license, last accessed: May 1st, 2025).

Table 4 summarizes the type of problems and their respective tested dimensions.

## G  ADDITIONAL IMPLEMENTATION DETAILS

### G.1  HARDWARE AND OPERATING SYSTEM

Due to the large number of benchmark problems and random seeds, the experiments are conducted in parallel on a distributed server with nodes of the same compute spec: a node with 22 Intel Xeon Platinum 8480+ CPUs cores and 1 NVIDIA H100 GPUs. All experiments were conducted on a GNU/Linux 6.5.0-15-generic x86_64 system running Ubuntu 22.04.3 LTS as the operating system, ensuring a consistent computational environment across all benchmark tests. As for the environment, we use BoTorch v0.12.0 and PyTorch 2.6.0+cu126 for all underlying optimization frameworks for the benchmark algorithms except GIT-BO.

Table 4: High-Dimensional Benchmark Problems

| Problems | Source | Type | Dimension ($D$) Tested |
|---|---|---|---|
| Ackley | Botorch (Balandat et al., 2020) | Synthetic | 100, 200, 300, 400, 500 |
| Dixon-Price | Botorch (Balandat et al., 2020) | Synthetic | 100, 200, 300, 400, 500 |
| Griewank | Botorch (Balandat et al., 2020) | Synthetic | 100, 200, 300, 400, 500 |
| Levy | Botorch (Balandat et al., 2020) | Synthetic | 100, 200, 300, 400, 500 |
| Michalewicz | Botorch (Balandat et al., 2020) | Synthetic | 100, 200, 300, 400, 500 |
| Powell | Botorch (Balandat et al., 2020) | Synthetic | 100, 200, 300, 400, 500 |
| Rastrigin | Botorch (Balandat et al., 2020) | Synthetic | 100, 200, 300, 400, 500 |
| Rosenbrock | Botorch (Balandat et al., 2020) | Synthetic | 100, 200, 300, 400, 500 |
| Styblinski-Tang | Botorch (Balandat et al., 2020) | Synthetic | 100, 200, 300, 400, 500 |
| Reactive Power Phase | CEC2020 Benchmark Suite (Kumar et al., 2020) | Real-World | 118 |
| Active Power (AP) Loss | CEC2020 Benchmark Suite (Kumar et al., 2020) | Real-World | 153 |
| Reactive Power Loss | CEC2020 Benchmark Suite (Kumar et al., 2020) | Real-World | 158 |
| Power Flow AP | CEC2020 Benchmark Suite (Kumar et al., 2020) | Real-World | 126 |
| Power Fuel Cost | CEC2020 Benchmark Suite (Kumar et al., 2020) | Real-World | 126 |
| Power AP+Fuel | CEC2020 Benchmark Suite (Kumar et al., 2020) | Real-World | 126 |
| Rover | Previous BO studies (Wang et al., 2018) | Real-World | 100, 200, 300, 400, 500 |
| MOPTA08 CAR | Previous BO studies (Papenmeier et al., 2022) | Real-World | 124 |
| MAZDA | Mazda Car Bechmark (Kohira et al., 2018) | Real-World | 222 |
| MAZDA SCA | Mazda Car Bechmark (Kohira et al., 2018) | Real-World | 148 |
| Walker Policy | Mujuco (Todorov et al., 2012; Wang et al., 2020) | Real-World | 102 |

## G.2 GIT-BO ALGORITHM IMPLEMENTATION DETAILS

The GIT-BO algorithm was implemented using Python 3.12 with the TabPFN v2.0.6 implementation and model (link: `https://github.com/PriorLabs/TabPFN` and `https://huggingface.co/Prior-Labs/TabPFN-v2-reg`, license: Prior Lab License (a derivative of the Apache 2.0 license (`http://www.apache.org/licenses/`))).

**Would making TabPFN differentiable hurt the performance?** Since there is no stable release of a TabPFN v2 code that allows full model differentiation as far as we know, we get rid of some marginal performance boosting numpy code in the official TabPFN v2 code (e.g., ensembling of 8 TabPFN v2 for increasing the accuracy marginally [4]) or rewrite the numpy-based operations (e.g., numerical transformations [5]) to PyTorch code into a single model TabPFN v2 in complete PyTorch code that allows us to use `torch.backward()` for gradient calculations. This change results in our implementation as faster inference speed due to the full GPU parallelization of using PyTorch and getting rid of the default `n_estimator=8` TabPFN v2 eight ensemble calculation (we use `n_estimator=1` with a fixed standardize transformation), but suffers from performance accuracy degradation as presented in Figure 13 without transformation permutations with ensembling. That said, the GIT-BO method would have even better performance if, in the future, TabPFN v2 releases a differentiable option.

## H LLM USAGE STATEMENT

We acknowledge the use of LLMs (ChatGPT, Claude, and Gemini) only for polishing the writing of this paper.

---

[4]https://github.com/PriorLabs/TabPFN/blob/main/src/tabpfn/preprocessing.py
[5]https://github.com/PriorLabs/TabPFN/blob/main/src/tabpfn/preprocessors/adaptive_quantile_transformer.py

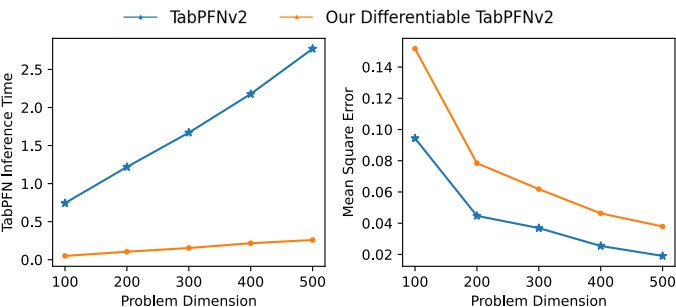

Figure 13: Comparison of TabPFN v2 and our implementation of GIT-BO TabPFN v2 across increasing problem dimensions. **Left:** inference time (seconds) grows substantially for TabPFN v2 due to ensemble evaluations, while GIT-BO's PyTorch implementation achieves consistent GPU-accelerated speedups. **Right:** mean squared error (MSE) highlights the accuracy trade-off, where eliminating TabPFN's default ensemble (n_estimator=8 → 1) leads to modest degradation. Overall, GIT-BO achieves faster inference with competitive accuracy, demonstrating the benefits of differentiable integration of TabPFN into BO pipelines.

