# OpenReview forum: "GIT-BO: High-Dimensional Bayesian Optimization with Tabular Foundation Models"
_ICLR.cc/2026/Conference — ICLR 2026 Poster_

### Official Review · Reviewer_hwuD · 2025-10-19

**Soundness:** 2
**Presentation:** 3
**Contribution:** 2
**Rating:** 2
**Confidence:** 4

**Summary:**

This paper introduces GIT-BO, a gradient-informed BO framework that aims to address the challenges of high-dimensional BO. The core idea is to integrate TabPFN v2, a tabular foundation model, as a surrogate model with a gradient-informed active subspace mechanism. This approach leverages the predictive-mean gradients from TabPFN v2 to identify a low-dimensional active subspace, aligning exploration to this subspace and using a UCB acquisition function without requiring online retraining. The authors claim that GIT-BO delivers a strong performance-time trade-off across 60 problem variants, including synthetic and real-world tasks, outperforming state-of-the-art GP-based methods.

**Strengths:**

- The core idea of integrating TabPFN v2 with gradient-informed active subspaces is an intuitive approach for handling the curse of dimensionality
- The paper highlights the potential for significant runtime advantages (orders-of-magnitude speedups) compared to GP-based methods, which is a crucial for real-world applications
- The paper presents extensive and thorough experimental results across a wide range of synthetic and real-world benchmarks

**Weaknesses:**

- All methods start with 200 LHS samples. For 500 total iterations, this means 40% of the budget is spent on initialization. Why such a large initialization? This heavily favors methods that converge quickly (like GIT-BO with fast TabPFN inference) over methods that need more iterations to refine their models (like SAASBO-NUT).
- Appendix B.1 (Figure 6) shows that vanilla TabPFN v2 without GI subspace performs poorly (8.6× worse regret). But this comparison is misleading. Vanilla TabPFN v2 uses random candidate sampling in the full 500D space, which is obviously inefficient. A fairer baseline would be TabPFN + random subspace projection or TabPFN + trust regions. Without these comparisons, it's unclear whether the gains come from gradient-informed subspace discovery specifically, or just from any dimensionality reduction paired with TabPFN.
- Appendix C (Figure 10) shows that GP+GI subspace performs poorly compared to TabPFN+GI. The authors hypothesize this is due to "poorly estimated GP hyperparameters" and "unreliable gradient estimates." However, in my eyes this suggests that the success is driven by TabPFN's quality as a surrogate, not by the GI subspace mechanism itself. If GI subspaces are theoretically principled (per Assumption 3), why do they fail with GPs? This casts doubt on the generality of the approach
- Beyond invoking "No Free Lunch," the paper does not deeply investigate why GIT-BO fails on Rover, Styblinski-Tang, etc. Are these problems outside TabPFN's pre-training distribution? Do they lack low-dimensional structure? A failure mode analysis would strengthen the contribution.

**Questions:**

- Can you provide empirical evidence that TabPFN's posterior approximation error $\epsilon_\text{approx}(t)$ actually vanishes (or remains small) on your benchmark tasks?
- The $\Phi$-Sobolev certificate in Li et al. (2024) applies to likelihood-informed subspaces in inverse problems, where the Fisher matrix is computed from a known likelihood model. In GIT-BO, you compute $H$ from TabPFN's predictive mean gradients, which are outputs of a learned, black-box transformer. Why is it valid to apply the same certificate?
- Why fix $r = 10$ for all problems? The authors mentioned in Appendix B.2 that adaptive variance-based selection (92.5%, 95%) achieves better average ranks than $r=10$, yet the main results use $r=10$

---

> ### Author Response · Authors · 2025-11-21
>
> We thank Reviewer hwuD for their thoughtful assessment and for highlighting several core strengths of our contribution. We appreciate the recognition of the intuitive integration of TabPFN with gradient-informed active subspaces, the importance of runtime improvements for real-world optimization, and the breadth and rigor of our empirical evaluation across synthetic and engineering benchmarks. We are also grateful that the reviewer found the presentation clear and the methodological framing well aligned with high-dimensional BO challenges. We aim to address each of the reviewer’s concerns—particularly regarding initialization size, baseline comparisons, GI-subspace robustness, and the behavior of TabPFN outside its pre-training distribution—with focused analyses and new experimental results, and we hope these clarifications will strengthen the reviewer’s overall assessment.
>
> ## Number of initial samples $N_{init}$
>
> We thank the reviewer for raising this point about initialization. To directly address it, we added a new experiment (Appendix B.7, Fig. 13) where we vary the initialization size across $N_{init}$ ∈ {20, 50, 200, 1000}, matching values used in prior high-dimensional BO work (20,50) [1] while addressing the question of large data regime from Reviewer RQPR (1000). For each setting, all algorithms are rerun 10 times and evaluated by average regret and statistical rank. **Across all regimes, GIT-BO maintains the best rank, while several GP baselines degrade or fluctuate**. Notably, smaller initializations favor TabPFN, consistent with prior PFN literature [2,3,4] that shows TabPFN performs well in low-data regimes. These results demonstrate that GIT-BO’s advantage is not an artifact of using $N_{init}$ = 200.
>
>
> [1] David Eriksson, Michael Pearce, Jacob Gardner, Ryan D Turner, and Matthias Poloczek. Scalable global optimization via local bayesian optimization. Advances in Neural Information Processing Systems, 32, 2019.
>
> [2] Noah Hollmann, Samuel Müller, Katharina Eggensperger, and Frank Hutter. Tabpfn: A transformer that solves small tabular classification problems in a second. arXiv preprint arXiv:2207.01848, 2022.
>
> [3] Noah Hollmann, Samuel Müller, Lennart Purucker, Arjun Krishnakumar, Max Körfer, Shi Bin Hoo, Robin Tibor Schirrmeister, and Frank Hutter. Accurate predictions on small data with a tabular foundation model. Nature, 637(8045):319–326, 2025.
>
> [4] Han-Jia Ye, Si-Yang Liu, and Wei-Lun Chao. A closer look at tabpfn v2: Understanding its strengths and extending its capabilities, 2025.
>
> ## Baselines for subspace reduction methods ablation
>
> We thank the reviewer for this suggestion. We address this concern by presenting an additional experiment in Appendix B.4, Fig. 10, that adds all requested baselines: **vanilla TabPFN, TabPFN + Trust Region (TR), TabPFN + BAxUS projection, and TabPFN + GI-subspace (GIT-BO)**. As a complementary analysis, Appendix B.5 (Fig. 11) further evaluates each subspace method under both EI and UCB, confirming that the relative ordering of subspace strategies is stable across acquisition functions. The revised results show that both TR and BAxUS projection improve over TabPFN, proving that simple random sampling in 500D without subspace restricted search space is indeed inefficient. However, **GI-subspace consistently matches or surpasses BAxUS projection on every synthetic and real-world benchmark**, while maintaining stronger final-iteration performance. These results indicate that the gains are not due to arbitrary dimensionality reduction but to gradient-informed subspace discovery, which remains the most effective strategy when paired with TabPFN.
>
> **We also note that TR and BAxUS interact less favorably with TabPFN because, unlike GP surrogates, TFMs lack kernel length-scales**—TR’s hypercube cannot be weighted by local length-scale geometry, reducing its effectiveness.
>
> ## Empirical Evidence for Vanishing Approximation Error
>
> We address the reviewer’s concern on error vanishing by adding an explicit experiment (Fig. 6) measuring the discrepancy between TabPFN’s predictive mean and a GP fitted on the same context set during BO, with the result presented in Appendix A.2 Figure 6. We can see that the average error across problems, the mean-squared error $ϵ_{approx}(t)$​ **decreases rapidly and approaches zero** as $t$ grows, providing direct empirical evidence that TabPFN’s posterior approximation error remains small in our benchmark regime.

---

> > ### Author Response · Authors · 2025-11-21
> >
> > ## Why did GI-subspaces initially fail for GPs originally, and what do updated results show?
> >
> > We thank the reviewer for bringing this observation to our attention. Our original GP+GI result used a non-BoTorch (Gpytorch) GP implementation and a custom gradient extractor, which we now recognize was suboptimal (https://docs.gpytorch.ai/en/stable/examples/08_Advanced_Usage/Simple_Batch_Mode_GP_Regression.html). In the revision, we adopt our recently discovered BoTorch’s propagate_grad (https://botorch.readthedocs.io/en/latest/settings.html) for getting gradient matrix and use the same GP configuration as Vanilla BO [1], ensuring reliable hyperparameter learning and fully consistent gradient computation. The updated results (Appendix C, Fig. 15) show that **GI-subspace identification generalizes to GPs as well**: the recovered intrinsic dimension converges to the correct ground-truth for most problems, while the regret curves approach the optimal value. These findings clarify that the earlier failure was implementation-related, and it is certainly not a flaw of our GI-subspace mechanism. Yet we highlight that this experiment does not affect our main contribution: GIT-BO’s integration of a frozen TFM with GI-guided exploration.
> >
> > [1] Carl Hvarfner, Erik Orm Hellsten, and Luigi Nardi. Vanilla Bayesian optimization performs great in high dimensions. In Proceedings of the 41st International Conference on Machine Learning, volume 235, pp. 20793–20817. PMLR, 2024.
> >
> > ## Failure-Mode Analysis for Rover / Styblinski–Tang
> >
> > Since **TabPFN’s pre-training tasks are not publicly released**, unfortunately we cannot directly determine whether Rover or StyblinskiTang fall within its prior distribution. Instead, we conducted a surrogate-diagnostic experiment measuring the normalized MSE (NMSE) at the true maximizer, which directly reflects whether the surrogate is reliable where BO decisions matter most. We prompt TabPFN with 10 set of different $X_{train}$ and  $X_{test}$ following our GIT-BO experiment’s sampling setup, get prediction on $X_{test}$ and calculate the NMSE at the true maximizer and take the average NMSE of the 10 trials. The result is shown in the table below. On engineering tasks (e.g., Active Power (AP) Loss, Reactive Power Loss, Power Flow AP, Car(x2) Mazda ) where GIT-BO performs well, TabPFN’s NMSE is smaller compared to the scalable synthetic functions and Rover exhibit relatively higher errors (0.35–0.52), indicating local miscalibration around the optimum. These synthetic problems intentionally contain no low-dimensional structure under our scaling, and Rover is multi-modal and sparse-reward, a regime where trust-region GP methods (e.g., TuRBO) are specifically designed to succeed. Thus, these tasks naturally make it challenging for GIT-BO. A deeper, model-specific failure-mode analysis is a promising direction for future work, but the main scope of this paper is to demonstrate that TFMs can serve as a competitive and computationally efficient surrogate for high-dimensional BO.
> >
> > | Problem Name | Dimension | Normalized MSE at the maximizer |
> > |---|---|---|
> > | Active Power (AP) Loss | 153D (unscalable) | *0.00679088* |
> > | Reactive Power Loss | 158D (unscalable) | *0.00907302* |
> > | Power Flow AP | 126D (unscalable) | *0.2609778* |
> > | Car(x2) Mazda | 148D (unscalable) | *0.10779438* |
> > | Griewank | 500D (scalable) | 0.48292013 |
> > | Michalewicz | 500D (scalable) | 0.52472899 |
> > | StyblinskiTang | 500D (scalable) | 0.45567671 |
> > | Rover | 500D (scalable) | 0.34518578 |
> >
> > ## Why fix $r$=10
> >
> > Our objective in this paper is to **evaluate GIT-BO in a strictly fair, out-of-the-box setting**. To avoid hyper-optimizing our own method while leaving all baselines exactly as implemented in BoTorch’s tutorial code, we intentionally refrain from tuning $r$. Although adaptive variance-based selection (92.5%, 95%) yields slightly better average ranks (Appendix B.1), using these tuned values only for GIT-BO while keeping all competing methods untuned would introduce an unfair advantage. We therefore fix $r$=10 across all benchmarks as a neutral, conservative choice aligned with our fairness philosophy. Importantly, even with this non-optimal hyperparameter, GIT-BO still achieves top-ranked optimization performance. If desired, we can update the central figure to include adaptive-$r$ results; GIT-BO remains superior under those settings as well.

---

> > > ### Author Response · Authors · 2025-11-21
> > >
> > > ## Theory on Fisher matrix $H$
> > >
> > > We thank Reviewer 4 for bringing this question to our attention. We address this question by updating Appendix A.2 Assumption 3 to incorporate the additional literature, thereby supporting our work.
> > >
> > > In summary, literature [1,2] have shown that using gradients coming from complex learned models or stochastic simulators for calculating empirical or approximate Fisher matrices can capture informative curvature and sensitivity structure for the development of gradient descent optimization algorithms. Moreover, second-order optimizers in the *Shampoo* and *SOAP* family and large-scale natural-gradient methods construct preconditioners directly from empirical gradient outer products or Kronecker-factored curvature estimates instead of an exact likelihood Fisher [3,4,5,6,7,8]. These works collectively show that **such approximations preserve the dominant eigenstructure needed for scaling and feature reweighting during training**. Likewise, Fisher-based sensitivity and dimension-reduction methods in engineering and Bayesian inverse problems use Fisher matrices estimated from Monte Carlo simulations or surrogate models to identify high-information directions [9]. Therefore, our use of TabPFN predictive-mean gradients to form an empirical Fisher matrix in GIT-BO follows this established practice of using approximate Fisher constructions to detect likelihood-informed, high-sensitivity subspaces.
> > >
> > > [1] Razvan Pascanu and Yoshua Bengio. Revisiting natural gradient for deep networks. arXiv preprint arXiv:1301.3584, 2013.
> > >
> > > [2] Frederik Kunstner, Philipp Hennig, and Lukas Balles. Limitations of the empirical fisher approximation for natural gradient descent. Advances in neural information processing systems, 32, 2019.
> > >
> > > [3] Runa Eschenhagen, Aaron Defazio, Tsung-Hsien Lee, Richard E. Turner, and Hao-Jun Michael Shi. Purifying shampoo: Investigating shampoo’s heuristics by decomposing its preconditioner. In The Thirty-ninth Annual Conference on Neural Information Processing Systems, 2025.
> > >
> > > [4] Vyas, Nikhil, Depen Morwani, Rosie Zhao, Mujin Kwun, Itai Shapira, David Brandfonbrener, Lucas Janson, and Sham Kakade. "Soap: Improving and stabilizing shampoo using adam." The Thirteenth International Conference on Learning Representations, 2025.
> > >
> > > [5] Yanqing Lu, Letao Wang, and Jinbo Liu. "Understanding SOAP from the Perspective of Gradient Whitening." arXiv preprint arXiv:2509.22938 (2025).
> > >
> > > [6] Jeffrey Pennington and Pratik Worah. The spectrum of the fisher information matrix of a single-hidden-layer neural network. Advances in neural information processing systems, 31, 2018.
> > >
> > > [7] Ryo Karakida, Shotaro Akaho, and Shun-ichi Amari. Universal statistics of fisher information in deep neural networks: Mean field approach. In The 22nd International Conference on Artificial Intelligence and Statistics, pp. 1032–1041. PMLR, 2019.
> > >
> > > [8] Osawa, Kazuki, Shigang Li, and Torsten Hoefler. "Pipefisher: Efficient training of large language models using pipelining and fisher information matrices." Proceedings of Machine Learning and Systems 5 (2023): 708-727.
> > >
> > > [9] Zahm, Olivier, Tiangang Cui, Kody Law, Alessio Spantini, and Youssef Marzouk. "Certified dimension reduction in nonlinear Bayesian inverse problems." Mathematics of Computation 91, no. 336 (2022): 1789-1835.
> > >
> > > ## Overview of updates since submission
> > >
> > > In response to reviewer feedback, we made several clarifications and added new analyses:
> > >
> > > - Appendix A.2 now includes a clearer explanation of Assumption 3.
> > > - Figure 6 provides an empirical verification of Assumption 1, as requested by Reviewer hwuD.
> > > - Appendix B.4–B.8 (Figs. 10–14) adds a suite of ablations addressing reviewer questions related to: subspace dimension choice, acquisition functions, reference point selection, initialization size, and finetuning.
> > > - Appendix C and Fig. 15 now include improved GP implementations, demonstrating that the GI-subspace mechanism also benefits GP-based BO, further supporting the generality of our approach.
> > >
> > > We have incorporated these new experiments into the updated manuscript pdf. We hope that the rebuttal and the new experiment provided in response to other reviewers will convince the reviewer to raise their score.

---

> ### Comment · Reviewer_hwuD · 2025-11-22
> **Reply to Authors' Rebuttal**
>
> I have read the rebuttal, and I am glad to see that the authors have satisfactorily addressed my concerns. I appreciate their effort in clarifying key aspects of the method and, more importantly, conducting additional experiments within the limited time frame. The new analyses have strengthened the paper significantly. I am happy to increase my score.

---

> > ### Author Response · Authors · 2025-11-24
> >
> > Thank you for your timely response and acknowledging the rebuttal points. We are happy to see that you were satisfied with our responses.

---

### Official Review · Reviewer_S5Jo · 2025-10-24

**Soundness:** 3
**Presentation:** 3
**Contribution:** 3
**Rating:** 6
**Confidence:** 4

**Summary:**

This paper proposes GIT-BO, a Bayesian optimization (BO) algorithm for high-dimensional black-box problems. GIT-BO uses prior fitted networks (PFNs) as the surrogate model, which allows for fast inference, where the popular Gaussian process surrogate has cubic inference complexity. While TabPFNv2, a tabular foundation model, already allows for modeling problems with up to 500 dimensions, the authors argue that additional steps are necessary to perform BO efficiently on high-dimensional problems. To this end, they use the gradient information provided by the PFN to estimate an active subspace. Like other subspace-based approaches for high-dimensional Bayesian optimization, they then sample points in the active subspace and project them to the full-dimensional space to evaluate the acquisition function and choose the next point to evaluate.

By comparing to several state-of-the-art algorithms for high-dimensional BO on various benchmarks, the authors provide extensive empirical evidence for the effectiveness of the proposed approach.

**Strengths:**

The paper addresses a relevant problem. High-dimensional BO has received considerable attention in the past and is an active field of research. The paper is the first method that uses PFNs, which is an interesting surrogate model due to its in-context capabilities, for high-dimensional BO.
The approach is well-motivated, and the paper is well-written. The storyline is clear, and the paper features an extensive empirical evaluation that shows the benefits of the approach. The evaluation is open about the limitations of GIT-BO and the performance of other state-of-the-art methods. The paper identifies the superior inference time of GIT-BO as a key strength, which is reasonable.
The paper provides an appendix with extensive ablation studies and additional experiments. The appendix complements the paper and preempts many questions I had when reading the paper.

**Weaknesses:**

The main concerns I have with this paper are the large performance degradations upon minor modifications of the algorithm. For instance, Figure 6 shows that the GIT-BO with expected improvement instead of the upper confidence bound performs considerably worse, worse than TabPFN without the gradient-informed subspace. Similarly, Figure 9 shows that the technique for sampling candidates in the low-dimensional subspace is crucial for performance. I wouldn’t expect such a big impact from these choices, and it makes me question the generalizability of the method.

**Questions:**

-	GIT-BO currently uses a search space of fixed dimensionality. Do you think the method would benefit from expanding subspaces, similar to BAxUS?
-	The reference point is centered on the centroid of the observed data. What is the impact of that choice? What happens if you center it on the incumbent observation, and why is this choice necessary?
-	Why does EI perform so poorly in Figure 6?
-	What is the difference between the sampling schemes in section B.4? What’s the difference between random sampling in the subspace and uniform sampling? And do you have any insight into why one performs better than the other?

---

> ### Author Response · Authors · 2025-11-21
>
> We thank Reviewer S5Jo for recommending the paper for acceptance and providing a thoughtful review, and for clearly highlighting the main strengths of our work. We appreciate the recognition that GIT-BO tackles an important high-dimensional BO problem and that the paper is well-motivated, clearly written, and empirically thorough. We also thank the reviewer for noting the novelty of applying PFNs in this setting, the breadth of our benchmarking, and the usefulness of the appendix in clarifying technical details. We are glad the reviewer identified the fast inference time of GIT-BO as a key advantage of the approach. Below, we address the reviewer’s questions and concerns in detail.
>
> ## Clarification on Subspace Expansion Question
>
> We thank the reviewer for raising this insightful question. Based on the BAxUS paper [1], BAxUS performs well essentially because (i) its nested random subspaces expand over time and (ii) it frees the user from specifying a fixed intrinsic dimension. In contrast, **GIT-BO’s subspace is not produced by random projections**: it is derived from the Fisher (gradient-covariance) structure of TabPFN, and therefore represents curvature-aligned directions rather than arbitrary embeddings. Because of this fundamental difference, directly concatenating BAxUS-style expansions with GI-subspaces is non-trivial and beyond the scope of this work; we view this as an interesting direction for future work. We again highlight that in this paper, **we aim to evaluate GIT-BO in a strictly fair, out-of-the-box setting, demonstrating the effectiveness of TFM as the BO surrogate and the GI-subspace**.
>
> Regarding the concern about fixed dimensionality, Appendix B.1 shows that GIT-BO already supports adaptive expansion or shrinkage through a variance-explained criterion. The $r$ selected by the 92.5–95% thresholds expands when the Fisher spectrum is rich and contracts when gradients concentrate, as visualized in Figure 15 on three problems. This adaptive strategy yields the best overall performance in Table 1, outperforming all fixed-r settings.
>
> Finally, we include in Appendix B.4 an ablation study that tests the concatenation of vanilla TabPFN-v2 + BAxUS’s projection and benchmarks against vanilla TabPFN-v2 (no subspace finding), TabPFN-v2 + Trust Region, and TabPFN-v2 + GI-subspace (GIT-BO). **While TabPFN-v2 + BAxUS’s projection improves convergence compared to vanilla TabPFN-v2, our GIT-BO methods still outperform it in both synthetic and engineering test problems**. We hope this addresses the reviewer’s concern.
>
> [1] Leonard Papenmeier, Luigi Nardi, and Matthias Poloczek. Increasing the scope as you learn: Adaptive bayesian optimization in nested subspaces. Advances in Neural Information Processing Systems, 35:11586–11601, 2022.
>
> ## Choice of reference point
>
> We thank the reviewer for raising this question. To evaluate the impact of the reference point, we added an ablation comparing two choices used in prior high-dimensional BO: centering the GI-subspace at the centroid of all observations $x_{ref}$ = $\bar{x}$ versus the incumbent best observation $x_{ref}$ = $x_{argmax{y_{obs}}}$ (used by [1,2]). As shown in Appendix B.6 and Figure 12, **centering at the centroid of all observations consistently yields lower regret and faster convergence** across all six benchmarks. We therefore adopt the centroid as the default in GIT-BO.
>
> We hypothesize that the centroid offers a more stable anchor by avoiding over-concentration around a possibly noisy incumbent and by keeping candidate generation within the well-sampled region where TabPFN’s in-context predictions are most reliable, aligned with PFN locality and imbalance analyses in recent work [3,4].
>
> [1] David Eriksson, Michael Pearce, Jacob Gardner, Ryan D Turner, and Matthias Poloczek. Scalable global optimization via local Bayesian optimization. Advances in neural information processing systems, 32, 2019.
>
> [2] Leonard Papenmeier, Luigi Nardi, and Matthias Poloczek. Increasing the scope as you learn: Adaptive bayesian optimization in nested subspaces. Advances in Neural Information Processing Systems, 35:11586–11601, 2022.
>
> [3] Han-Jia Ye, Si-Yang Liu, and Wei-Lun Chao. A closer look at tabpfn v2: Understanding its strengths and extending its capabilities, 2025.
>
> [4] Ismail Nejjar, Faez Ahmed, and Olga Fink. Im-context: In-context learning for imbalanced regression tasks. Transactions on Machine Learning Research, 2024.

---

> > ### Comment · Reviewer_S5Jo · 2025-11-24
> >
> > I would like to thank the authors for their detailed rebuttal.
> >
> > I have one concern regarding the reference point. The **majority of the synthetic benchmarks** used in the paper (Ackley, Dixon-Proce, Levy, Rastrigin, Powell) have to optimum at the origin. Methods like BAxUS can exploit this property, and, arguably, the authors did not run synthetic benchmarks with the optimum in the origin for that exact reason.
> >
> > I wonder how far the authors ensure that choosing the reference point in the origin does not have similar effects for GIT-BO, considering that most benchmark problems benefit from that choice.

---

> > > ### Author Response · Authors · 2025-11-27
> > >
> > > ## Clarifications on the concerns for synthetic problems’ origin-centered optima
> > > We thank Reviewer S5Jo for reading our rebuttal and raising this insightful question. We address the concern about synthetic problems’ origin-centered optima in three points:
> > > 1. **Clarification on benchmark composition**: Only 4 of 9 synthetic functions (Ackley, Griewank, Powell, Rastrigin) have origin-centered optima at $(0,0,...,0)^D$. Other synthetic functions have optima that are not centered at the origin (e.g., Dixon-Price's optimum lies at $x_i = 2^{-(1-2^{-(i-1)})}$ and Rosenbrock at $(1,1,...,1)^D$). Taken across all 60 problem variants, this means well **under half of the synthetic cases, and about one‑third of all problems,** including real‑world tasks, **are origin‑centered**. Our suite deliberately mixes landscapes to avoid bias.
> > > 2. **GIT-BO does not center at the origin**: Our reference point is **$x_{\text{ref}} = \bar{x}_{\text{obs}}$, the centroid of observed samples, not the domain center**. This data-driven anchor tracks the optimization trajectory and does not systematically favor origin-centered optima. In contrast, BAxUS constructs sparse projections where $S^\top \mathbf{0} = \mathbf{0}$, guaranteeing the origin remains in the embedded subspace. Our gradient‑informed basis is data‑dependent, so the search space is centered where evidence accumulates rather than around the origin.
> > > 3. **Empirical validation**: Following BAxUS's own robustness experiment (Papenmeier et al., 2022) to investigate the difference between the BAXUS and GIT-BO’s projection method, we shift five synthetic problems with optima at $(0,0,...,0)^D$ (Ackley, Griewank, Powell, Rastrigin) or $(1,1,...,1)^D$ (Rosenbrock) by uniform offsets $\delta_i \sim U(x^{\text{LB}}, x^{\text{UB}})$. The result of this empirical study is shown in Appendix B.9, Figure 15 in our updated manuscript pdf. GIT-BO consistently outperforms BAxUS across all five shifted benchmarks, confirming that our gradient-informed mechanism does not rely on an origin-centered structure.
> > >
> > > We hope this clarifies why GIT‑BO is not advantaged on origin‑centered problems. We are happy to add a short note in the camera‑ready to make this fairness point explicit.

---

> > > > ### Comment · Reviewer_S5Jo · 2025-11-27
> > > >
> > > > Thank you for the clarification and the thorough rebuttal. I found that all my concerns were addressed, and I will increase my score.

---

> > > > > ### Author Response · Authors · 2025-11-27
> > > > >
> > > > > We sincerely appreciate your careful evaluation and are glad that our response addressed the concerns. Thank you for your favorable assessment.

---

> ### Author Response · Authors · 2025-11-21
>
> ## GIT-BO’s UCB vs. EI
>
> We thank the reviewer for raising this point. In revisiting our EI implementation, we identified a small implementation inconsistency in how the GI-subspace step was applied in several EI runs. To ensure complete clarity and methodological consistency, we re-ran the full UCB vs. EI ablation using a fully verified and cross-checked evaluation pipeline.
>
> The updated results in Appendix B.5 (Figure 11) show a clear and consistent pattern across all benchmarks: both EI and UCB achieve substantially more substantial convergence when paired with our gradient-informed subspace, outperforming alternative projection strategies such as trust region search (TR) or BAxUS-style embeddings. Importantly, UCB provides a modest but stable advantage over EI across the six problems presented, which motivates our choice of UCB as the default acquisition function in GIT-BO. This aligns with the empirical and theoretical findings on EI’s improvement term, which becomes numerically unstable in high dimensions due to its probability-of-improvement factor collapsing as variance concentrates, whereas UCB maintains well-behaved exploration [1,2].
>
> These updated results address the reviewer’s concern by (i) confirming that **GIT-BO is not dependent on a specific acquisition rule** and (ii) demonstrating that the **GI-subspace mechanism generalizes effectively across acquisition functions**. We hope this clarification helps reinforce confidence in our design choices.
>
> [1] Sebastian Ament, Samuel Daulton, David Eriksson, Maximilian Balandat, and Eytan Bakshy. Unexpected improvements to expected improvement for bayesian optimization. Advances in Neural Information Processing Systems, 36:20577–20612, 2023.
>
> [2] Zhitong Xu, Haitao Wang, Jeff M. Phillips, and Shandian Zhe. Standard gaussian process is all you need for high-dimensional bayesian optimization. In The Thirteenth International Conference on Learning Representations, 2025.
>
> ## Clarification on the sampling schemes
>
> Our random sampling draws i.i.d. Gaussian vectors (np.random.randn) projected through $V_r$, while uniform sampling draws from the hypercube (numpy.random.uniform). Sobol sampling uses low-discrepancy sequences (torch.SobolEngine) that provide more even coverage. Empirically (Appendix B.3 Fig. 9), uniform sampling performs well on synthetic functions with broad, smooth basins. **We hypothesize that it explores the full GI-box more aggressively**. In contrast, Gaussian and Sobol sampling yield slightly better convergence on the narrower, more structured engineering objectives since they concentrate the samples near the centroid and provide denser local coverage. **These differences appear problem-dependent rather than universally optimal**. A full study of adaptive GI-subspace sampling strategies is beyond the scope of this paper, but represents a promising direction for future work.
>
> ## Overview of updates since submission
>
> In response to reviewer feedback, we made several clarifications and added new analyses:
>
> - **Appendix A.2** now includes a clearer explanation of Assumption 3.
> - **Figure 6** provides an empirical verification of Assumption 1, as requested by Reviewer hwuD.
> - **Appendix B.4–B.8 (Figs. 10–14)** adds a suite of ablations addressing reviewer questions related to: subspace dimension choice, acquisition functions, reference point selection, initialization size, and finetuning.
> - **Appendix C** and **Fig. 15** now include improved GP implementations, demonstrating that the GI-subspace mechanism also benefits GP-based BO, further supporting the generality of our approach.
>
> We have incorporated these new experiments into the updated manuscript pdf. We hope that the rebuttal and the new experiment provided in response to other reviewers will convince the reviewer to raise their score.

---

### Official Review · Reviewer_RQPR · 2025-10-29

**Soundness:** 3
**Presentation:** 3
**Contribution:** 3
**Rating:** 6
**Confidence:** 5

**Summary:**

The paper presents a novel method GIT-BO that utilises the TabPFN foundational model as a surrogate, and uses its gradients to identify a low-dimensional subspace over which to conduct the acquisition optimisation. Authors benchmark the time and sample complexity of their method against other baselines and also conduct a number of ablations showing the importance of each of the components of the final algorithm.

**Strengths:**

- The proposed method is faster (in terms of wall-clock time, while maintaining performance comparable to the existing state of art
- The authors conduct plenty of interesting ablations, showing the importance of each of the components used in the final algorithm
- The additional theory in the appendix, while not particularly novel and easily following from preceding work, is still a nice addition for completeness

**Weaknesses:**

- Since authors emphasise the importance of time-complexity, as opposed to pure sample complexity as it is typically done in BO literature, it would be nice to demonstrate a problem setting, where we actually care about time-complexity (e.g. high-throughput BO), as in most classical BO problems, sample complexity is paramount, whereas wallclock time is of secondary importance
- It seems to be authors focused on a relative low-data regime, where fitting a GP is still relatively fast, it would be more interesting to see what happens when the size of the dataset is much bigger, where technically fitting a GP should be slow, yet TabPFN should still be fast

**Questions:**

- In Appendix B1, Figure 6, you show that replacing UCB with EI in your method makes the performance drop drastically on Levy and Rosenbrock. This is quite unprecedented. Can you provide some explanation as to why that happens?

---

> ### Author Response · Authors · 2025-11-21
>
> We thank Reviewer RQPR for recommending paper acceptance and providing a thoughtful assessment, and for recognizing the efficiency of GIT-BO by leveraging TabPFN as a surrogate together with gradient-informed subspace discovery. We appreciate the reviewer’s positive remarks on the extensive ablations, the theoretical additions in the appendix, and the method’s strong time-complexity performance relative to existing BO approaches. We also value the reviewer’s careful identification of limitations, particularly regarding the high-throughput motivation and the behavior of UCB versus EI, and view these as important opportunities to further clarify the scope and empirical properties of GIT-BO. We hope that the explanations and new experiments provided in the rebuttal will address these concerns and lead the reviewer to consider increasing their score.
>
> ## Clarification on the Time-Complexity Matters in Modern BO
>
> We thank the reviewer for highlighting the importance of clarifying when wall-clock runtime becomes the primary bottleneck. In fact, several **recent work across top AI venues explicitly focuses on fast Bayesian optimization**. For example, Aaron et al. [1] accelerate HPO on large datasets by optimizing learning-curve–based partial evaluations, and Buathong \& Frazier [2] introduce fast BO for function networks with partial evaluation to reduce computational load per iteration. Runtime constraints also directly impact benchmarking methodology: Papenmeier et al.'s BAxUS [3] shows that several BO algorithms cannot be evaluated for the same iteration count because runtime becomes the limiting factor, making fair comparisons impossible. Therefore, **we follow Xu et al.'s work [4] and record CPU/GPU time per iteration** as a primary performance metric.
>
> **Beyond AI, engineering domains provide even stronger evidence that compute time is often the dominant practical constraint**. Siemenn er al.'s ZoMBI [5] demonstrates up to 400× computational speedups for materials discovery, where slow BO would prohibit real-time experimental steering. In robotics, automation, and self-driving laboratories, runtime directly limits experimental throughput: **autonomous experimental researchers, self-driving laboratories, and even NSF-funded physical-AI initiatives** all emphasize that human-/robot-in-the-loop optimization must operate at interactive speeds [6,7,8].
>
> Yu et al.'s work [9] in BO for engineering design clearly states that the automobile manufacturing process and active experimentation with human-in-the-loop BO require fast BO. In this context, **we believe that our results on engineering design tasks --- including the Mazda and MOPTA vehicle problems, power systems, and robotics control --- demonstrate that GIT-BO meets these fast-decision constraints**. As shown in Figure 4, GIT-BO consistently achieves strong performance across these time-sensitive settings.
>
> [1] Klein, Aaron, Stefan Falkner, Simon Bartels, Philipp Hennig, and Frank Hutter. "Fast bayesian optimization of machine learning hyperparameters on large datasets." In Artificial intelligence and statistics, pp. 528-536. PMLR, 2017.
>
> [2] Buathong, Poompol, and Peter I. Frazier. "Fast Bayesian Optimization of Function Networks with Partial Evaluations." arXiv preprint arXiv:2506.11456 (2025).
>
> [3] Leonard Papenmeier, Luigi Nardi, and Matthias Poloczek. Increasing the scope as you learn: Adaptive bayesian optimization in nested subspaces. Advances in Neural Information Processing Systems, 35:11586–11601, 2022.
>
> [4] Zhitong Xu, Haitao Wang, Jeff M. Phillips, and Shandian Zhe. Standard gaussian process is all you need for high-dimensional bayesian optimization. In The Thirteenth International Conference on Learning Representations, 2025.
>
> [5] Siemenn, Alexander E., Zekun Ren, Qianxiao Li, and Tonio Buonassisi. "Fast Bayesian optimization of Needle-in-a-Haystack problems using zooming memory-based initialization (ZoMBI)." npj Computational Materials 9, no. 1 (2023): 79.
>
> [6] Gongora, Aldair E., Bowen Xu, Wyatt Perry, Chika Okoye, Patrick Riley, Kristofer G. Reyes, Elise F. Morgan, and Keith A. Brown. "A Bayesian experimental autonomous researcher for mechanical design." Science advances 6, no. 15 (2020): eaaz1708.
>
> [7] Adesiji, Adedire, Jiashuo Wang, Cheng-Shu Kuo, and Keith A. Brown. "Benchmarking Self-Driving Labs." Digital Discovery (2025).
>
> [8] Lauren Marlier. \$2M NSF Grant for Self-Driving Labs Will Accelerate Discovery. URL: https://cbe.ncsu.edu/nsf-grant-self-driving-labs/.
>
> [9] Rosen Ting-Ying Yu, Cyril Picard, and Faez Ahmed. Fast and accurate Bayesian optimization with pre-trained transformers for constrained engineering problems. Structural and Multidisciplinary Optimization, 68(3):66, 2025.

---

> ### Author Response · Authors · 2025-11-21
>
> ## Low-data Regime
>
> We thank the reviewer for raising the question of performance in larger-data regimes, where GP-based BO becomes slow while TabPFN-based inference should remain fast. To directly address this, we added a new experiment in Appendix B.7 (Fig. 13) that expands the data regime by varying the initialization size $N_{init}$ in {20, 50, 200, 1000}, with $N_{init}$ = 1000 representing a dataset 5× larger than that used in the main paper. This experiment simultaneously resolves Reviewer hwuD’s concern about the sensitivity of GIT-BO to small initial samples (even lower data regime).
>
> **Across all initialization sizes, including the largest regime where re-fitting GP surrogates is computationally taxing, GIT-BO maintains the best statistical rank**, while several GP-based baselines degrade or fluctuate considerably. These results empirically validate that GIT-BO continues to perform reliably even when the dataset becomes substantially larger and the GP competitors slow down. Finally, we acknowledge that TabPFN inherits a practical upper limit from its fixed context window (TabPFN v2 has a limit of 10k rows and 500 feature columns); scaling beyond this limit would require larger TFMs or advanced ensembling strategies, which we identify as promising directions for future work.
>
>
> ## GIT-BO’s UCB vs. EI
>
> We thank the reviewer for raising this point. In revisiting our EI implementation, we identified a small implementation inconsistency in how the GI-subspace step was applied in several EI runs. To ensure complete clarity and methodological consistency, we re-ran the full UCB vs. EI ablation using a fully verified and cross-checked evaluation pipeline.
>
> The updated results in Appendix B.5 (Figure 11) show a clear and consistent pattern across all benchmarks: both EI and UCB achieve substantially more substantial convergence when paired with our gradient-informed subspace, outperforming alternative projection strategies such as trust region search (TR) or BAxUS-style embeddings. Importantly, UCB provides a modest but stable advantage over EI across the six problems presented, which motivates our choice of UCB as the default acquisition function in GIT-BO. This aligns with the empirical and theoretical findings on EI’s improvement term, which becomes numerically unstable in high dimensions due to its probability-of-improvement factor collapsing as variance concentrates, whereas UCB maintains well-behaved exploration [1,2].
>
> These updated results address the reviewer’s concern by (i) confirming that **GIT-BO is not dependent on a specific acquisition rule** and (ii) demonstrating that the **GI-subspace mechanism generalizes effectively across acquisition functions**. We hope this clarification helps reinforce confidence in our design choices.
>
> [1] Sebastian Ament, Samuel Daulton, David Eriksson, Maximilian Balandat, and Eytan Bakshy. Unexpected improvements to expected improvement for bayesian optimization. Advances in Neural Information Processing Systems, 36:20577–20612, 2023.
>
> [2] Zhitong Xu, Haitao Wang, Jeff M. Phillips, and Shandian Zhe. Standard gaussian process is all you need for high-dimensional bayesian optimization. In The Thirteenth International Conference on Learning Representations, 2025.
>
> ## Overview of updates since submission
>
> In response to reviewer feedback, we made several clarifications and added new analyses:
>
> - **Appendix A.2** now includes a clearer explanation of Assumption 3.
> - **Figure 6** provides an empirical verification of Assumption 1, as requested by Reviewer hwuD.
> - **Appendix B.4–B.8 (Figs. 10–14)** adds a suite of ablations addressing reviewer questions related to: subspace dimension choice, acquisition functions, reference point selection, initialization size, and finetuning.
> - **Appendix C** and **Fig. 15** now include improved GP implementations, demonstrating that the GI-subspace mechanism also benefits GP-based BO, further supporting the generality of our approach.
>
> We have incorporated these new experiments into the updated manuscript pdf. We hope that the rebuttal and the new experiment provided in response to other reviewers will convince the reviewer to raise their score.

---

> > ### Comment · Reviewer_RQPR · 2025-11-26
> >
> > Thank you for your clarifications. I remain positive about the paper.

---

> > > ### Author Response · Authors · 2025-11-27
> > >
> > > We thank the reviewer for the additional consideration and for the positive assessment.

---

### Official Review · Reviewer_MZj9 · 2025-11-01

**Soundness:** 3
**Presentation:** 3
**Contribution:** 3
**Rating:** 6
**Confidence:** 4

**Summary:**

This paper introduces GIT-BO, a gradient-informed Bayesian optimization framework that leverages TabPFN v2, a tabular foundation model, to perform high-dimensional Bayesian optimization without surrogate retraining. The key idea is to extract predictive-mean gradients from the frozen TabPFN model, identify a low-dimensional active subspace via Fisher information, and perform UCB-based acquisition within this subspace. The method is evaluated on 60 high-dimensional benchmark problems (up to 500D), demonstrating strong performance and runtime advantages over GP-based baselines like SAASBO, TuRBO, and BAXUS.

**Strengths:**

++ This method combines frozen tabular foundation models (TFMs) with gradient-informed subspace discovery, and provides a novel fusion of amortized inference and classical dimension reduction.

++ Comprehensive benchmarking against SOTA methods, with rigorous statistical ranking and runtime analysis.

**Weaknesses:**

-- The performance depends on the pre-trained foundation model. The frozen TFM may not adapt well to functions outside its pre-training distribution, leading to poor performance on certain tasks.
Additionally, no fine-tuning or domain adaptation is performed, which limits generalization to highly specialized or out-of-distribution objectives.

**Questions:**

Please see the weakness.

---

> ### Author Response · Authors · 2025-11-21
>
> We thank Reviewer MZj9 for recommending paper acceptance and providing a thoughtful review of our method, GIT-BO, and for recognizing the novelty of combining frozen tabular foundation models with gradient-informed subspace discovery. We are grateful for the acknowledgement of our comprehensive benchmarking across 60 high-dimensional problems and the rigorous runtime and statistical analyses. We also appreciate the reviewer’s clear articulation of the primary concern regarding potential limitations of frozen TFMs on highly specialized or out-of-distribution objectives.
>
> ## Clarification on TFM finetuning:
>
> Our design choice to not finetune TabPFN in the main paper is intentional: we aim to evaluate GIT-BO under a strictly fair, out-of-the-box setting, where all baseline algorithms and surrogate models, including TabPFN, are used without task-specific hyperparameter tuning. This isolates the algorithmic contribution of GIT-BO and reflects the core advantage of TFMs: strong zero-shot performance without retraining. Finetuning is an orthogonal enhancement rather than a requirement.
>
> Notably, **recent work consistently shows that continued pre-training or light finetuning improves TabPFN-style models when domain shift is present** [1,2,3,4]. Therefore, we perform a finetuning experiment to test whether this hypothesis holds and whether GIT-BO’s performance can be enhanced by finetuning.
>
> To directly address the reviewer’s concern, we incorporated a finetuning experiment into the updated manuscript (Appendix B.8, Fig. 14). We finetuned TabPFN on each benchmark using 1,000 problem-specific samples, following the continued-pretraining procedures proposed in Real-TabPFN [1,2]. **As reported in Figure 14, finetuning yields uniform but modest performance improvements**, confirming the findings of [1,2,3,4]. Importantly, the frozen TFM already performs strongly across most tasks, and our main result and conclusion on GIT-BO’s top ranking would not be changed based on this additional finding.
>
> These results demonstrate that:
> - **Finetuning helps** when the distribution shift is significant, and
> - **The frozen model is sufficiently robust** for GIT-BO in high-dimensional optimization, supporting our main evaluation strategy.
>
>
> [1] Anurag Garg, Muhammad Ali, Noah Hollmann, Lennart Purucker, Samuel Muller, and Frank Hutter. Real-tabpfn: Improving tabular foundation models via continued pre-training with real-world data. arXiv preprint arXiv:2507.03971, 2025.
>
> [2] Léo Grinsztajn, Klemens Flöge, Oscar Key, Felix Birkel, Philipp Jund, Brendan Roof, Benjamin Jäger, Dominik Safaric, Simone Alessi, Adrian Hayler, et al. Tabpfn-2.5: Advancing the state of the art in tabular foundation models. arXiv preprint arXiv:2511.08667, 2025.
>
> [3] Junwei Ma, Valentin Thomas, Rasa Hosseinzadeh, Alex Labach, Jesse C. Cresswell, Keyvan Golestan, Guangwei Yu, Anthony L. Caterini, and Maksims Volkovs. TabDPT: Scaling tabular foundation models on real data. In The Thirty-ninth Annual Conference on Neural Information Processing Systems, 2025.
>
> [4] Joshua P Gardner, Juan Carlos Perdomo, and Ludwig Schmidt. Large scale transfer learning for tabular data via language modeling. In The Thirty-eighth Annual Conference on Neural Information Processing Systems, 2024.
>
> ## Overview of updates since submission
>
> In response to reviewer feedback, we made several clarifications and added new analyses:
>
> - **Appendix A.2** now includes a clearer explanation of Assumption 3.
> - **Figure 6** provides an empirical verification of Assumption 1, as requested by Reviewer hwuD.
> - **Appendix B.4–B.8 (Figs. 10–14)** adds a suite of ablations addressing reviewer questions related to: subspace dimension choice, acquisition functions, reference point selection, initialization size, and finetuning.
> - **Appendix C** and **Fig. 15** now include improved GP implementations, demonstrating that the GI-subspace mechanism also benefits GP-based BO, further supporting the generality of our approach.
>
> We have incorporated new experiments into the updated manuscript pdf. We hope that the rebuttal and the new experiment provided in response to other reviewers will convince the reviewer to raise their score.

---

> > ### Comment · Reviewer_MZj9 · 2025-11-26
> > **Reply to Authors' Rebuttal**
> >
> > Thanks for the author's response. I have no other questions and would like to maintain my positive rating.

---

> > > ### Author Response · Authors · 2025-11-27
> > >
> > > We thank the reviewer for the additional consideration and for the positive assessment.

---

### Author Response · Authors · 2025-11-21
**Summary of our rebuttal**

We thank all reviewers for their thoughtful feedback. Three reviewers recommend acceptance and highlight the novelty, clarity, and thoroughness of GIT-BO. Reviewer hwuD raises a few concerns, which we hope to have addressed below by making substantial clarifications, adding new experiments, and expanding several analyses across the main paper and appendix. Below, we provide a concise overview of all updates, organized by theme, to clearly illustrate how each concern has been addressed.

- **GI-subspace vs. alternatives (Appendix B.4)**
    - New baselines: TabPFN alone, TabPFN+Trust-Region, TabPFN+BAxUS projection.
    - **Result:** GIT-BO (TabPFN+GI-subspace) is consistently strongest, showing gains are not due to arbitrary dimensionality reduction.
- **Acquisition functions (UCB vs. EI) (Appendix B.5)**
    - After fixing an implementation bug, both EI and UCB improve substantially with a GI-subspace.
    - **Result:** UCB is modestly more stable in high-dimensional setting (Fig. 8), addressing reviewer concerns about EI underperformance.
- **Sampling strategies & reference point (Appendix B.2, B.6)**
    - Added ablations on uniform/Gaussian/Sobol sampling and centroid vs incumbent as reference point.
    - **Result:** Uniform sampling + centroid is consistently more stable.
- **Initialization size (20 to 1000 points) (Appendix B.7)**
    - **Result:** GIT-BO remains top-ranked in all regimes; GP baselines degrade or fluctuate (Fig. 13).
- **Finetuning TabPFN (Appendix B.8)**
    - **Result:** Finetuning yields modest uniform gains; frozen-model GIT-BO already achieves top performance.
- **GI-subspace with GP surrogates (Appendix C, Fig. 15)**
    - Using improved BoTorch gradient extraction, GI-subspaces also benefit GPs.
    - **Result:** Confirms GI-subspaces are not TFM-specific.
- **Failure-mode analysis (Rover, Styblinski–Tang)**
    - We added a surrogate miscalibration experiment to understand cases where the model doesn't work very well.
    - **Result:** Rover and hard scalable functions show significantly higher error, giving evidence towards possible failures because of OOD behavior or lack of exploitable low-D structure.
- **Stronger theoretical grounding (Appendix A.2, Fig. 6)**
    - We clarified Assumption 3, explicitly linked the empirical Fisher construction to certified Fisher-eigenstructure literature, and added an experiment verifying Assumption 1.
    - **Result:** Fig. 6 shows that TabPFN’s posterior mean converges toward the GP posterior as context grows, providing direct empirical support for surrogate fidelity.

The revision directly addresses every substantive reviewer's concern with new evidence, not just explanation. The new experiments demonstrate that:

- GIT-BO’s gains are not fragile and persist across acquisition functions, sampling choices, subspace sizes, initialization sizes, and even surrogate type (GP vs TabPFN).
- The GI-subspace is the critical mechanism, not random projection or TFM bias.
- The theoretical assumptions are now better motivated and empirically validated.
- The few failure cases arise from potential surrogate miscalibration, not methodological flaws.

---

### Author Response · Authors · 2025-12-03
**Note to the Incoming Area Chair**

Dear AC,

Given the recent reverting of scores to their pre‑discussion state, we wanted to briefly summarize how the reviewers’ assessments evolved after our rebuttal and the new experiments and analyses described in our earlier **“Summary of our rebuttal”** comment.

- The initial scores were 6, 6, 6, 2, with the 2 driven by concerns about initialization size, fairness of baselines, the generalizability of the GI‑subspace mechanism, and the need for stronger theoretical and empirical support.
- In response, we updated the manuscript with additional experiments and clarifications in **Appendix B and C**, as described in our earlier **“Summary of our rebuttal”** comment.
  - Updates include: additional ablations on subspace methods and acquisition functions, initialization‑size study, finetuning experiment, updated GP+GI results, origin‑shift robustness tests, and a surrogate failure‑mode analysis, along with expanded theoretical discussion.
- After reading these updates,
  - Reviewer hwuD wrote that our new analyses had **“strengthened the paper significantly”** and explicitly stated that they were **happy to increase their score**.
  - Reviewer S5Jo wrote that all concerns were addressed and that they would **increase their score**.
  - Reviewers MZj9 and RQPR both confirmed that they **remained positive** about the paper.

Before the incident, the reviewers’ numeric scores had already been updated from 6, 6, 6, 2 to **6, 6, 8, 6**.

We fully understand that only the pre‑discussion scores are now considered official, and we respect the program chairs’ decision. We mention this only to help reconstruct the state of the discussion and the reviewers’ final views after the rebuttal and additional experiments, in case it is useful for your assessment.

Thank you very much for your time and for handling this process under difficult circumstances.

Best regards,
The Authors

---

### Meta-Review · Area_Chair_bPNS · 2026-01-10

**Summary:**

The following concerns were raised:

- Concerns about the large initial sample size.
- Lack of depth/analysis or conjecture explaining e.g. the handful of worse results, or just generally other failure modes of the method.
- Fairness to baselines, especially in high dimensions.
- Concerns that the TFN was not fine tuned.

**Reviewer Concerns:**

I think the reviewers do a good job of addressing all of these concerns, particularly in their response to Reviewer hwuD. I generally think in places where new results were not supplied the responses were reasonable (e.g., the authors' comments about not finetuning the TFN).

**Reviewer Scores:**

All of the reviewers clearly indicated in comments that they would increase their score, suggesting that their concerns were addressed.

---

### Decision · Program_Chairs · 2026-01-26

Accept (Poster)